



# A Framework for Parameter Estimation, Sensitivity Analysis, and Uncertainty Analysis for Holistic Hydrologic Modeling Using SWAT+

Salam A. Abbas[1], Ryan T. Bailey[1], Jeremy T. White[2], Jeffrey G. Arnold[3], Michael J. White[3], Natalja
Cerkasova[4], Jungang Gao[4]

[1] Department of Civil and Environmental Engineering, Colorado State University, USA.
[2] INTERA, Inc. Perth Australia.
[3] Grassland Soil and Water Research Laboratory, USDA–ARS, Temple, TX, 76502, USA.
[4] Blackland Research & Extension Center, Texas A&M AgriLife, Temple, TX, 76502, USA.

*Correspondence to*: Salam A. Abbas (salam.a.abbas@colostate.edu)

**Abstract.** Parameter Sensitivity analysis plays a critical role in efficiently determining main parameters, enhancing the effectiveness of estimation of parameters, and uncertainty quantification in hydrologic modeling. In this paper, we demonstrate uncertainty and sensitivity analysis technique for the holistic SWAT+ model, coupled with new *gwflow* module, spatially distributed, physically based groundwater flow modeling. Main calculated groundwater inflows and outflows
include boundary exchange, pumping, saturation excess flow, groundwater–surface water exchange, recharge, groundwater–lake exchange, and tile drainage outflow. We present the method for four watersheds located in different areas of the United States for 16 years (2000–2015), emphasizing regions of extensive tile drainage (Winnebago River, Minnesota, Iowa), intensive surface–groundwater interaction (Nanticoke River, Delaware, Maryland), groundwater pumping for irrigation (Cache River, Missouri, Arkansas), and mountain snowmelt (Arkansas Headwaters, Colorado).
The main parameters of coupled SWAT+*gwflow* model are estimated utilizing the parameter estimation software (PEST). The monthly streamflow of holistic SWAT+*gwflow* is evaluated based Nash–Sutcliffe efficiency index (NSE), percentage bias (PBIAS), determination coefficient ($R^2$), and Kling–Gupta efficiency coefficient (KGE), whereas groundwater head is evaluated using mean absolute error (MAE). The Morris method is employed to identify the key parameters influencing hydrological fluxes. Furthermore, the iterative ensemble smoother (iES) is utilized as a technique for Uncertainty
Quantification (UQ) and Parameter Estimation (PE) and to decrease the computational cost owing to the large number of parameters.

Depending on the watershed, key identified selected parameters include aquifer specific yield, aquifer hydraulic conductivity, recharge delay, streambed thickness, streambed hydraulic conductivity, area of groundwater inflow to tile, depth of tiles below ground surface, hydraulic conductivity of the drain perimeter, river depth (for groundwater flow
processes); runoff curve number (for surface runoff processes); plant uptake compensation factor, soil evaporation compensation factor (for Potential and actual evapotranspiration processes); soil available water capacity, percolation



coefficient (for Soil water processes). The presence of *gwflow* parameters permits for the recognition of all key parameters in the surface/subsurface flow processes, with results substantially differing if the base SWAT+ models are utilized.

**Keywords:**

model calibration; SWAT+; *gwflow*; parameter sensitivity; Morris screening; uncertainty quantification; iterative ensemble smoothers.

## 1. Introduction

Hydrologic models have been developed to enhance understanding of the dynamics of hydrological fluxes to address practical issues related to water resources management (Liu et al., 2020; Wei et al., 2018), especially under the influence of

anthropogenic activities and climate change, which can result in significant changes in the hydrological system (Abbas et al., 2022; Pokhrel et al., 2021). Typically, hydrologic models include several parameters to represent the hydrologic processes and to consider spatial variations resulting from climate, soil type, land use, etc. (Fatichi et al., 2016; Čerkasova et al., 2021). To employ hydrologic models in a responsible manner for system understanding and scenario analysis, sensitivity analysis (SA), uncertainty analysis (UA), and parameter estimation (PE) are key steps in the modeling process due to the presence of

spatial heterogeneities (Bennett et al., 2013; Doherty and Hunt, 2009) and often the use of a broad suite of model parameters. SA identifies model parameters that have a strong influence on model output (e.g., streamflow), and results generally can provide insights into system behavior and point to system parameters that require more data collection or management strategies that may be efficient in controlling a certain system response (Leta et al., 2015). UA relates uncertainty in model parameters to model output, and hence can provide ranges of system output possibilities, e.g., when using the model in

scenario analysis as a decision support tool, to answer questions regarding effects of system changes. PE provides the best values for matching model predictions to historical observations.

SA methods can be classified into local sensitivity analysis (LSA) and global sensitivity analysis (GSA) (Santos et al., 2022). Examples of LSA approaches are one–variable–at–a–time (OAT) and the differential analysis (DA) method (Devak and Dhanya, 2017), are less reputable since they disregard to consider the interaction between several parameters and cannot

precisely estimate optimal parameters value (Helton, 1993). While GSA techniques such as regional sensitivity analysis (RSA), Morris screening, variance-based sensitivity analysis (Sobol's method), and Fourier amplitude sensitivity test (FAST), have been developed and used in many applications (Olaya–Abril et al., 2017; Devak and Dhanya, 2017). These methods take into account the interaction between different parameters by altering several parameters of model together (Pianosi et al., 2017; Devak and Dhanya, 2017). GSA is gaining prominence in hydrologic and environmental modeling

(e.g., Plischke et al., 2013; Pianosi et al., 2017). GSA is employed for the detection of insignificant parameters and the identification of influential parameters with a significant impact on model outputs (Santos et al., 2022).



Other GSA applications include identification of model behavior, prioritization for uncertainty estimation and reduction, and for simplification of the model (Pianosi et al., 2017). However, these methods typically require a large number of model evaluations. More recently, iterative ensemble smoother (iES) techniques have been developed for uncertainty quantification

(UQ) and for more efficient parameter estimations (PE) by reducing the number of model evaluations incurred by large number of parameters (Chen and Oliver, 2012); this technique can be implemented in a non-intrusive/model-independent approach, resulting in a desirable option for application to analyses of hydrologic and environmental modeling. The iES has been utilized in several applications (e.g., Bocquet and Sakov, 2014; Crestani et al., 2013).

Although SA-UA-PE methods have been applied numerous times to watershed models such as SWAT (Arnold et al., 1998)

(e.g., Pianosi et al., 2017; Nossent et al., 2011; Qiu et al., 2019), their application to coupled surface-subsurface models is sparse (e.g., Herzog et al., 2021; Wu et al., 2014; Ryken et al., 2020). For example, the coupled SWAT-MODFLOW model (Bailey et al., 2016) has been applied to regions worldwide (e.g., Izady et al., 2022; Abbas et al., 2022; Sith et al., 2019); and more recently, the SWAT+ model (Bieger et al., 2017) with the *gwflow* module (Bailey et al., 2020) has been applied to simulate hydrological processes in watershed systems; but these models have been applied without SA and in a deterministic

manner, i.e., without including UA. In addition, PE has been challenging, with often SWAT and MODFLOW calibrated separately before being linked, which can be attributed to the complexity in the interaction between SWAT and MODFLOW, as well as the high dimensionality of the parameter space of these two models.

In this paper, we demonstrate the use of SA, PE, and UA methods in a coupled SWAT+ *gwflow* model to identify surface and subsurface parameters that control two key watershed responses: streamflow and groundwater head. Hydrologic fluxes

in the coupled model include vegetation ET, surface runoff, infiltration, soil percolation and recharge, saturation excess flow, groundwater-stream exchange, soil lateral flow, groundwater pumping, groundwater-lake exchange, tile drainage outflow, and boundary exchange. Targeted parameters include soil properties, evaporation parameters, runoff curve number, snow parameters, aquifer properties (hydraulic conductivity, specific yield), streambed properties (hydraulic conductivity, thickness), and tile drain parameters. The chosen SA method is the Morris screening method, joined to a PE method using

the PEST software program (Doherty, 2020). In an alternate method, we demonstrate the use of UA in the PE process, using an iterative ensemble smooth (iES) to establish prior and posterior ensembles of parameters and system responses. Both methods (PE-SA; iES) can be key components in the application of coupled surface-subsurface models to watershed systems. While in this paper we demonstrate methods for the SWAT+ *gwflow* modeling system, they can be applied to other hydrologic models.

We demonstrate the methods for four 8-digit watersheds throughout the conterminous United States: Nanticoke River (Delaware, Maryland), Arkansas Headwaters River (Colorado), Winnebago River (Minnesota, Iowa), and Cache River (Missouri, Arkansas). These watersheds are chosen owing to distinct hydrologic characteristics, such as snowmelt dominant basin (Arkansas Headwaters), shallow groundwater (Nanticoke), the extensive networks of subsurface tile drains (Winnebago), and groundwater pumping for irrigation (Cache). The SWAT+*gwflow* models were simulated for each

watershed from 2000 to 2015, with a two-year warm-up period (2000–2001), seven-year calibration period (2002–2008), and





seven-year testing period (2009–2015). These models were tested based on annual groundwater head and measured monthly streamflow measured at USGS monitoring wells and stream gages, correspondingly. Preliminary models of SWAT+ *gwflow* for the Winnebago River watershed, the Nanticoke River watershed, and the Cache River watershed were presented in Bailey et al. (2023), but only uncalibrated results were provided. This current study establishes possible SA-UA-PE methods

to increase model accuracy to a level suitable for scenario analysis (e.g., conservation practices, changes in climate and land use) in these watersheds.

## 2. Materials and Methods

### 2.1. Modeling framework for the study watersheds

Figure 1 presents four watersheds in United States with different hydrologic features were selected for SWAT+*gwflow*

simulation: Nanticoke River (Delaware, Maryland), Arkansas Headwaters River (Colorado), Winnebago River (Minnesota, Iowa), and Cache River (Missouri, Arkansas). A comprehensive summary of the primary characteristics of each watershed is presented in Table 1. The annual precipitation rates vary between 425 mm (Arkansas Headwaters) to 1,287 mm (Cache), while the total surface area of the watersheds varies considerably, from 1,787 km$^2$ for Winnebago to 7,940 km$^2$ for Arkansas Headwaters. Each watershed is a headwater 8-digit watershed and is in a different 2-digit region.

These four watersheds were specifically chosen on account of distinctive hydrologic characteristics that demonstrate informative application of the *gwflow*, such as: high baseflow with extensive groundwater discharge to streams (Nanticoke; Wolock, 2003), extensive presence of tile drainage (Winnebago), humid climate (Cache and Nanticoke), semi–arid climate (Arkansas Headwaters), extensive groundwater pumping for irrigation (Cache), and mountain snowmelt (Arkansas Headwaters). A detailed map of study areas showing watershed boundaries, streams, 12-digit catchment boundaries (i.e.,

subbasin), USGS river gage stations, USGS groundwater monitoring well locations, weather station locations, and water bodies, is shown in Fig. 2.

### 2.1.1. SWAT+ Model

The Soil and Water Assessment Tool (SWAT) (Arnold et al., 1998) is a process based, basin scale, semi–distributed, continuous–time hydrologic model that has been applied in many countries around the world for watershed management,

policy development, and environmental planning (Bieger et al., 2015; Zhang et al., 2020). The SWAT model was developed and designed by the Agricultural Research Service (ARS) of the United States Department of Agriculture (USDA) to simulate spatial and temporal variations in processes and fluxes of water, nutrients, and sediment. Common uses of the model include assessing water supply, nutrient loads, and sediments loads under historical and future conditions of climate, land use, and land management practices within watersheds and river basins of varying scale (e.g., Ghaffari et al., 2010;

Wang et al., 2018; Bhatta et al., 2019). The main computational unit within SWAT is the hydrologic response unit (HRU), unique geographic areas of soil type, land use, and topographic slope (Neitsch et al., 2011), with fluxes aggregated at the



subbasin level and then routed to streams. Stream routing occurs from upstream to downstream, with total watershed yield of water, nutrients, and sediment occurring at the watershed outlet.

The SWAT modeling code has recently been restructured to SWAT+ (Bieger et al., 2017), which provides additional
flexibility in routing water, nutrients, and sediment between watershed spatial objects (HRUs, aquifers, reservoirs, channels, routing units, wetlands). As an example, fluxes can be routed from HRU to HRU, or from channel to channel within a single subbasin, whereas the original SWAT only allowed routing from HRUs to channels, and each subbasin had a single channel. However, as with the original SWAT model, the groundwater processes are treated simplistically, assuming steady state conditions and homogeneous aquifers, and without physically based movement of groundwater and exchange with surface
water features using hydraulic head potential and differences. Hence, the *gwflow* module was created for SWAT+ to allow representation of groundwater processes and fluxes in a physically based manner (Bailey et al., 2020), as described in Section 2.1.2.

In this study, we use SWAT+ models that have been created within the National Agroecosystem Model (NAM) (White et al., 2022; Arnold et al., 2020), a national effort for improving environmental assessments and conservation strategies. Within the
NAM, a SWAT + model is constructed for each of the 2,139 HUC8 (8–digit hydrologic unit code) watersheds within the conterminous United States, simulating hydrologic processes and management according to five domains: main rivers (>150 km$^2$), tributaries (15–150 km$^2$), headwaters (1–15 km$^2$), transitions (0.2–2.0 km$^2$), and fields (1–50 ha). Table 2 lists the datasets used to create each SWAT+ model using publicly available data sources. Each cultivated field is designated as a unique HRU, with remaining HRUs delineated based on topographic slope, land use, and soil type. Subbasin boundaries
coincide with HUC12 catchments within each HUC8 watershed. Each NHD+ channel segment is designated as a unique channel in SWAT+. White et al. (2022) provides detailed information on model construction and input data sets. We use these model set-ups for the four study watersheds.

**2.1.2. *gwflow* Module**

The *gwflow* module (Bailey et al., 2020, 2022) is constructed and combined with SWAT+ for physically based spatially
distributed groundwater storage and flow modeling in unconfined aquifer systems, to replace the original SWAT+ groundwater module. The default SWAT+ groundwater module simulates groundwater fluxes with homogeneous aquifer properties, absence of groundwater flow between nearby aquifer systems, and groundwater discharge to streams based on aquifer storage and release parameters, as a substitute to distributed values of gradients and head differences. If the *gwflow* module is activated, the routine is called during each daily time step of the simulation. *gwflow* utilizes a set of grid cells to
simulate groundwater storage and flow through time (Fig. 3). Each grid cell has a specified aquifer volume, calculated using the ground surface elevation, bedrock elevation, and specific cell widths. Groundwater storage V (m$^3$) is updated during each daily time step (time *n* to time *n + 1*) for each cell (*i, j*) using a groundwater balance equation:


$$V_{i,j}^{n+1} = V_{i,j}^{n} + \left( sources_{i,j}^{n} - sinks_{i,j}^{n} \pm lateral\ flow_{i,j}^{n} \right) \left( t^{n+1} - t^{n} \right) \tag{1}$$

Sources consist of recharge, stream seepage, and lake seepage; sinks consist of groundwater ET, saturation excess flow, groundwater discharge to streams, pumping, tile drainage outflow, and groundwater discharge to lakes; and lateral flow

refers to Darcy flow between adjacent cells, based on cell-specific hydraulic conductivity ($K$) and head gradients. Recharge is provided from HRUs, using a geographic intersection between HRUs and grid cells. Groundwater-stream exchange, groundwater-lake exchange, and tile drainage outflow are calculated with Darcy's Law, using object properties (e.g., streambed conductivity, stream width, stream length). Groundwater pumping can be specified or simulated based on crop irrigation demand, conditioned on available groundwater storage. Once the new volume is calculated, a new value of head is

calculated using specific yield ($S_y$) of the grid cell. With the inclusion of the *gwflow* module, SWAT+ simulates land surface, soil, and channel processes, and the *gwflow* module simulates subsurface processes (Fig. 4), with several interface fluxes (soil recharge, saturation excess flow, groundwater-stream exchange, groundwater-lake exchange, tile drainage to streams). Cell size (m) for the Winnebago, Cache, and Nanticoke watersheds was set at 500 m, whereas cell size for the Arkansas Headwaters, due to a larger spatial extent of the watershed, was set at 1000 m (Table 1). Datasets used to populate *gwflow*

cell values (Table 2) include aquifer thickness (ground surface to bedrock; Fig. 5), geologic units for $K$ and $S_y$, locations of tile drainage, and USGS groundwater monitoring wells for initial groundwater head in the year 2000. For the latter, spatial interpolation is used between wells to provide a head value for each cell. Cells for groundwater-stream exchange and groundwater-lake exchange are identified by intersecting cells with NHD+ channels and water bodies (see Table 2, Fig. 2), respectively. We note that basic model set-up for the Winnebago, Nanticoke, and Cache watersheds is provided in Bailey et

al. (2023), in an initial demonstration of modifying SWAT+ models of the NAM to include the *gwflow* module.

### 2.2. SA-UA-PE Methods for the SWAT+ models

In this section, we describe the application of SA, UA, and PE tools to the watershed models constructed in Section 2.1. The general application of these tools to SWAT+ *gwflow* is summarized in the schematic of Fig. 6. In this study, we demonstrate two possible operations: 1) PE with PEST followed by SA with the Morris method, to identify system parameters that

control streamflow and groundwater head for each watershed; and 2) PE and UA with iES, to provide prior and posterior ensembles of parameters and system responses (streamflow). The next sections describe the individual tools, and how they are applied to the four watersheds.

### 2.2.1. Method #1: Parameter ESTimation Tool (PEST) followed by Sensitivity Analysis

The SWAT+*gwflow* models are constructed based on daily time step with 2 years warm-up period (2000–2001), for the

calibration period of 2002–2008, and validation period of 2009–2015. SWAT+*gwflow* models are first calibrated and tested using PEST (Doherty, 2020), a nonlinear, model-independent parameter estimator. PEST uses a local optimization technique that utilizes the Gauss-Marquardt-Levenberg algorithm (Doherty, 2004) to minimize the user-defined objective function



(e.g., minimization of root mean squares between simulated and observed values). PEST has been broadly employed for sensitivity analysis, uncertainty quantification, and model calibration for water quality and hydrologic models (e.g., Rode et al., 2007; Bahremand and De Smedt, 2010; Jiang et al., 2014).

In this study, we use all available monthly streamflow from USGS stream gage stations and average annual groundwater head from USGS monitoring wells in the objective function (OF). There are 1, 2, 3, and 4 stream gaging sites for the Winnebago, Nanticoke, Cache, and Arkansas Headwaters watersheds, respectively, and 7, 26, 92, and 3 monitoring wells (Fig. 2). The contribution of each of these sites to the composite OF were adjusted by manipulating the weights applied to the residuals to ensure that each site is of similar magnitude and significance in determining the optimal parameter values. Local optimization criterion (LOC) can be described as the weighted sum of OF. Objective function is computed as the squared sum of weighted residuals. LOC and OF can be expressed as:

$$OF = \sum_{j=1}^{n} \left[ x_{j,obs} - x_{j,sim} \right]^2 \tag{2}$$

$$LOC = \sum_{i=1}^{m} \omega_i \, OF_i \tag{3}$$

where $n$ is the total number of the measured/simulated streamflow or groundwater monitoring wells, $m$ is the total number of the observation groups of the observed streamflow from the gaging stations and groundwater monitoring wells, and $\omega$ is the weight of the related objective function.

The monthly simulated streamflow of SWAT+$gwflow$ models of the four study watersheds is evaluated using determination coefficient ($R^2$), Nash–Sutcliffe Efficiency Index (NSE), Kling–Gupta Efficiency Index (KGE), and percent of bias (PBIAS). The mean absolute error (MAE) is used to evaluate performance of groundwater level at USGS monitoring wells.

Based on SWAT model literature (e.g., Arnold et al., 2013; Koo et al., 2020), we selected 23 parameters to be modified by PEST (Table 3), focusing on surface runoff, evaporation, soil properties, groundwater processes, and snowmelt accumulation and melt processes. We set 2000–2001 as the warm-up period, 2002-2008 as the calibration period, and 2009-2015 as the testing period. Therefore, in the initial PEST runs, we only use simulation periods of 2000-2008. Once PEST is finished for each watershed model, we then run each model for 2000–2015 to quantify criteria results (i.e., NSE, $R^2$, PBIAS, KGE, and MAE).

Once a parameter set was established using PEST, we applied the Morris screening method to each model to assess the impact of each parameter on streamflow and groundwater head. Morris Screening (Elementary Effects Test) (Morris, 1991) is a qualitative global sensitivity analysis (GSA) technique that computes the relative sensitivity of model parameters, by calculating the change in the model output given a change in the model parameter $x_i$ value (i.e., elementary effect), with all other parameter values held constant. This procedure occurs over a range of parameter values, yielding a relationship between the parameter value and the model output. The following equation demonstrates the computation of a single elementary effect for the $i^{th}$ parameter:





$$EE_i = \frac{f(x_1, \ldots, x_i + \Delta_i, \ldots, x_p) - f(x)}{\Delta_i} \tag{4}$$

where $EE_i$ is the elementary effect value of the $i^{th}$ model parameter, $f$ represents the model, $x_1, \cdots, x_i$ is the model parameter value, and $\Delta$ represents the change. Within this method, the mean $\mu$ and standard deviation $\sigma$ of all $EE_i$ for a parameter are often used to assess the sensitivity or significance of parameters. To prevent the canceling of positive and negative values of 220 $EE_i$, Campolongo et al. (2007) proposed using the absolute value of $EE_i$, yielding the mean $\mu^*$. Therefore, $\mu^*$ and $\sigma$ can be calculated as follows for a given parameter $x_i$:

$$\mu_i^* = \frac{1}{n} \sum_{j=1}^{n} |EE_i(j)| \tag{5}$$

$$\sigma_i = \sqrt{\frac{1}{n-1} \sum_{j=1}^{n} \left[ EE_i(j) - \frac{1}{n} \sum_{j=1}^{n} EE_i(j) \right]^2} \tag{6}$$

where $n$ is the number of $EE_i$ computations. The $\mu^*$ for the model parameters are then ranked, to determine the parameters that have the strongest influence on model output. In this study, we implemented the Morris method using the software tool pestpp-sen (White et al., 2020), variation of PEST. Table 4 lists parameters and their ranges for the four study watersheds. 225 The number of classes (column 3) refers to the number of unique zones or categories for each parameter. For example, for Winnebago River, there are 4 aquifer zones, each with a different value of $K$ and $S_y$.

**2.2.2. Method #2: Iterative Ensemble Smoother (iES) for Parameter Estimation and UA**

In a second method, we use an iES (Chen and Oliver, 2013) to establish prior and posterior uncertainty estimates of model parameters, within the pestpp-ies (White, 2018) framework that uses the PEST model interface. The iES is based on the 230 original Ensemble Kalman Filter (EnKF) (Evensen, 1994), a data assimilation algorithm that updates state variables through assimilation of measured data into model results, based on correlations between the state variables and the measurement data. For model parameters that have a strong influence on model results, the parameter values can also be updated through this data assimilation. Updates to state variables and parameters occur in a sequence of update steps. The EnKF was implemented in a "smoother" scheme, the Ensemble Smooth (ES) (Van Leeuwen and Evensen, 1996), in which all past 235 states and parameters are updated in a single update step, using all past measurement data. Chen and Oliver (2013) modified the ES to perform iteratively using the Gauss-Levenberg-Marquardt (GLM) algorithm, resulting in a significant decrease in computational burden for models with many parameters.

The iES method starts with an initial ensemble of values for each parameter (i.e., a "prior" ensemble). An estimation to a Jacobian matrix of parameter sensitivities is computed based on the relationships between model parameters and model 240 output, using a range of parameter values based on the prior parameter ensemble (Chen and Oliver, 2013). The contents of the Jacobian matrix are then used to update the ensemble of each model parameter, by seeking to minimize model residuals



using the GLM algorithm. The result of the process is a posterior ensemble of model parameters, that are optimally consistent with measured data. Table 4 lists parameters and their ranges used for iES application to the four study watersheds with 3 iterations of the data assimilation algorithm (250 model runs) in pestpp-ies.

## 3. Results and Discussion

We first present hydrologic results for each of the four study watersheds through application of PEST, followed by the results of the Morris sensitivity analysis and the iES application.

### 3.1. Hydrologic State Variables and Fluxes

### 3.1.1. Streamflow and General Water Balance

The comparison between observed and simulated monthly streamflow at 10 locations showed a good model performance based on NSE, $R^2$, PBIAS, and KGE as presented in Fig. 7 and Table 5 which shows the hydrograph of observed and simulated streamflow at four selected gages. By utilizing desktop computer, an Intel® Core ™ i7-10700 CPU @ 2.90 GHz with 64 GB RAM, simulation times for whole period of simulation (2000–2015) for the four watersheds with SWAT+ and SWAT+*gwflow* are presented in Table 6, ranges (3–13) minutes for base SWAT+ and (7–35) minutes for SWAT+*gwflow*. The simulation time of these models is relatively faster than other explicitly physically based integrated hydrologic models.

### 3.1.2. General Watershed Fluxes

Table 7 displays the annual average hydrologic fluxes for the four study watersheds. Catchment key inflows include groundwater inflow from adjacent aquifer along the catchment boundary and precipitation. Catchment key outputs comprise soil lateral flow, surface runoff, evapotranspiration ET, tile drainage flow, saturation excess flow, and stream seepage.

The internal flows to the watershed include surface water irrigation (calculated by SWAT+), pumping irrigation (computed by *gwflow*), recharge (computed by *gwflow*), and groundwater-reservoir/lake exchange (calculated by *gwflow*). Table 7 also reveals key hydrologic fractions and average annual water yield. Cache has an annual value of (141 mm) for groundwater pumping for irrigation. Notably, Winnebago has the highest flow of tile drain (62 mm), Nanticoke River demonstrates high fluxes of groundwater to the stream network with saturation excess flow of (183 mm).

Arkansas Headwaters and Cache have small net groundwater discharge to stream (+ 37 Sat excess flow −1.7 mm seepage = + 35.3 mm for cache) and (– 4 mm seepage + 4.6 Sat excess flow = + 0.6 mm for Arkansas Headwaters), owing to deeper groundwater levels in comparison to stream stage. The baseflow contribution is moderate (> 0.30) for Nanticoke River, and low (< 0.20) for the other three watersheds. The yield fraction, i.e., the ratio of water yield in the streams to precipitation) ranges from 0.19 (Arkansas Headwaters) to 0.48 (Nanticoke). The recharge fraction ranges from 0.01 (Arkansas Headwaters) to 0.08 (Winnebago), with recharge fluxes for several of the watersheds similar in magnitude to soil lateral flow and surface runoff.





### 3.1.3. Monthly Hydrologic Fluxes

Figure 8 reveals monthly hydrologic flow processes for the period of (2002–2015) for each watershed. Plots on the left show results for the entire watershed system, whereas plots on the right show results for the aquifer system. Key watershed inflows are (boundary inflow and precipitation), where watershed outflows are (tile drainage, groundwater saturation excess flow, runoff, surface ET, and lateral flow) that are showed watershed seasonal fluxes for each basin. The Winnebago River is notable for its high flux rates of tile drainage outflow, groundwater exchange with reservoirs/lakes in the Arkansas Headwaters River, seasonal pattern of saturation excess flow (i.e., groundwater that reaches the river due to groundwater flooding) in the Nanticoke River, and groundwater pumping in the Cache River watershed, exhibiting the unique hydrologic characteristics of each watershed in relation to groundwater storage and flow.

### 3.1.4. Groundwater Head

Figure 9 contains the statistical performance based on mean absolute error (MAE) of annual groundwater level for four study watersheds for the period of 2000–2015. MAE results show an acceptable error (< 1.5 m residual in groundwater level) between simulated and measured average annual groundwater head at each USGS monitoring well site. However, a few locations have higher error (2.5– 3.6 m difference), although these residuals are small compared to the saturated thickness of the aquifer.

### 3.1.5. Spatial Variation of Groundwater Fluxes

Figure 10 shows saturated thickness maps (that is vertical distance between bedrock and water table) for the final year of simulation (2015) for the study watersheds, with saturated thickness similar in spatial pattern to the thickness of the unconfined aquifer (see Figure 5) but differing due to spatial changes in groundwater head within each watershed.

Raster maps of average daily groundwater sink/source flow processes (Fig. 11, 12, and 13) demonstrate zones of stress within the aquifer unit and regions of main inflows into the stream channel system. Spatial fluxes of recharge, groundwater-stream interaction (i.e., saturation excess flow), and groundwater pumping are presented as maps in Fig. 11, Fig. 12, and Fig. 13, respectively. Saturation excess flow occurs where the water table is shallow. Groundwater pumping for irrigation is presented for Nanticoke and Cache, since the other two watersheds do not experience groundwater pumping for irrigation. Cache has the highest pumping rates, due to extensive irrigation practices in the region.

### 3.2. Sensitivity Analysis using the Morris Screening Method

The Morris results for parameter influence on streamflow (Fig. 14) show the most influential parameters for each study watershed:

1) For Winnebago: percolation coefficient (Perco1), streambed thickness (bed_thick), hydraulic conductivity of the drain perimeter (tile_k), and streambed hydraulic conductivity (bed_k). These results indicate that streamflow is





controlled principally by processes that affect tile drainage and stream-aquifer interactions. This is somewhat surprising, as surface runoff is the dominant flux contributing to streamflow.

2) For Nanticoke: specific yield (syaqu2), hydraulic conductivity (kaqu2), streambed thickness (bed_thick), and streambed hydraulic conductivity (bed_k). These results indicate that groundwater properties and processes control streamflow, in agreement with the high baseflow fraction (0.32) of the watershed (Table 7).

3) For Arkansas Headwaters: Melt factor for snow on June 21 (Mmax), Snowmelt base temperature (Mtmp), streambed thickness (bed_thick), Snowpack temperature lag factor (Tmplag), and curve number (cn_frstgd). This is not surprising, as the streamflow is dominated by spring-time snowmelt patterns.

4) For Cache: Soil evaporation compensation factor (esco), percolation coefficient (Perco2), specific yield (syaqu4), streambed hydraulic conductivity (bed_k), curve number (rcsr_gd), available water capacity (awc3), thickness (bed_thick), Plant uptake compensation factor (epco), and recharge delay (rech_del). Streamflow in this watershed is dominated by processes that affect surface runoff (421 mm in Table 7) and groundwater pumping (141 mm).

Morris results for parameter influence on groundwater level (Fig. 15) show the most influential parameters for each study watershed:

1) For Winnebago: streambed hydraulic conductivity (bed_k), indicating the strong influence of stream-aquifer interactions on groundwater head in the region.

2) For Nanticoke: specific yield (syaqu2) and hydraulic conductivity (kaqu2).

3) For Arkansas Headwaters: hydraulic conductivity (kaqu9) and specific yield (syaqu6).

For Cache: Soil evaporation compensation factor (esco), curve number (rcsr_gd), and available water capacity (awc3 and awc4), indicating the influence of land surface and soil processes on groundwater head, due to their control on the volume of groundwater that is pumped from the aquifer.

The estimated time-varying parameter sensitivity calculated by the Morris method are represented in Fig. 16 for the most influential parameters in the four watersheds. The results for Winnebago River reveal that (bed_thick) and (perco1) are significantly increased with time indicating that the distribution of soil water capacity may alter. For Nanticoke River, (syqau2) and (kaqu2) varies drastically over the period of simulation. Similar behavior observed for other parameters (Mmax, Mtmp, esco, bed_k), but with less fluctuation.

### 3.3. Uncertainty Analysis and Parameter Estimation using the iES

Figure 17 shows the observed and best estimated monthly streamflow with prior and posterior prediction uncertainty band for the four study watersheds. The plots in the left column represent prior parameter ensembles (uncalibrated Monte Carlo results) with wide uncertainty bands. Meanwhile, the plots in the right column show the posterior ensemble that effectively reduces the uncertainty band. For example, in Arkansas Headwaters, the prior ensemble uncertainty band was shifted to the





left of the measured streamflow, owing to an incorrect characterization of snowmelt timing and magnitude. However, the posterior ensemble uncertainty band is much narrower and fits the timing and magnitude of the measured streamflow.

Figure 18 demonstrates the effect of data assimilation on the parameters more quantitatively, which compares the histogram of prior parameter ensembles (gray), with the histogram of posterior parameter ensembles (blue), for 9 of the most influential parameters in the four study watersheds. The posterior distribution of parameters is narrower than the prior distribution, which helps in the estimation of model parameters. The range parameters for curve number–Cache River (Fig. 18–H) and specific yield–Nanticoke River (Fig. 18–C) indicate the largest influence of data assimilation. Short correlation ranges have

been reduced from the posterior.

Figure 19 shows the influence of data assimilation on the average annual water balance more quantitatively, which compares the histogram of prior ensembles (gray), with the histogram of posterior ensembles (blue), for 8 most important water balance components in the four study watersheds. The posterior distribution of parameters is narrower than the prior distribution, which helps in the estimation of water balance component.

In general, the application of the iES can provide ensembles of posterior parameter sets that, when used in the model, provide simulation results that are in close comparison with measured data. And, due to the use of ensembles, includes uncertainty in results. When used for scenario analysis and decision making, these models can employ the posterior ensembles of parameters to propagate uncertainty into model results, therefore serving effectively in the role of decision support.

**4. Summary and Conclusion**

In this article, we present two methods to include sensitivity analysis, uncertainty analysis, and parameter optimization into coupled surface-subsurface hydrologic models, using the SWAT+ model as an example. The method utilizes the *gwflow* module, which is a spatially distributed, physically based groundwater flow module coupled to the SWAT+ model, which utilizes aquifer control volumes (i.e., grid cells) to compute daily water balance in an unconfined aquifer. We present our

technique for four different U.S. watersheds: Winnebago River, Nanticoke River, Cache River, and Arkansas Headwaters. These watersheds were selected on account of their respective unique hydrologic features: an extensive network of tile drain (Winnebago), shallow groundwater (Nanticoke), snow-melt dominant (Arkansas Headwaters), and extensive groundwater pumping for irrigation (Cache).

The SWAT+*gwflow* models are calibrated based on the monthly streamflow and annual groundwater level for the period of

2000–2008 with 2-years warm-up period, validated for a period of 2009–2015. The parameter estimation software (PEST) and (PEST++) are used for the calibration, sensitivity analysis, and uncertainty analysis of hydrologic models. Additionally, watershed water balance fluxes are evaluated for stability of models. All watershed models showed good statistical performance of streamflow simulation (10 River gages locations) and groundwater level results (128 monitoring wells), however, a few wells exhibited high values of mean absolute error results. Model outputs comprising saturated thickness





(spatial maps), raster maps of groundwater flow processes (saturation excess flow, stream seepage, pumping, recharge) which can be utilized to validate the model and recognize areas that need further parameter estimation, groundwater head (time series and spatial maps of observation locations), and stream discharge. By combining average annual water balance fluxes, groundwater head, and streamflow data, hydrologic flow processes can be restricted to realistic ranges. Increased fidelity in process representation allows these modeling tools to be utilized for the assessment of water resources under

different land use and climate scenarios over a wide range of hydrologic conditions.

GSA using Morris screening technique was applied to SWAT+*gwflow* models of study watersheds to assess the governing system factors on surface runoff and groundwater fluxes. The pestpp-sen tool within the PEST++ environment is utilized to generate parameter values, update model files for SWAT+*gwflow* models, run the model simulations, and compute sensitivity indices for the Morris method. The sensitivity of 23 parameters (including surface runoff fluxes, actual and

potential evapotranspiration fluxes, groundwater flow fluxes, snow fluxes, and soil water fluxes) were investigated based on 2 model responses: minimizing monthly streamflow and minimizing the mean absolute error (MAE) of annual groundwater head data.

The iES method was used for the model input uncertainty for the prior (uncalibrated results) and posterior ensembles, thus, resulting in better uncertainty prediction that will improve the utilize of hydrologic models in decision-making. This

technique is implemented using pestpp-ies tool within the PEST++ environment.

From the results we conclude that:

1) Winnebago River (extensive presence of tile drainage): groundwater flow-related parameters and soil water parameters significantly affect streamflow and groundwater heads, especially percolation coefficient, streambed thickness, hydraulic conductivity of the drain perimeter, and streambed hydraulic conductivity.

2) Nanticoke River (intensive surface–groundwater interaction): groundwater flow-related parameters notably influence streamflow and groundwater heads, specifically specific yield, hydraulic conductivity, streambed thickness, and streambed hydraulic conductivity.

3) Arkansas Headwaters River (snowmelt dominant basin): Snow processes and surface runoff flow related parameters extensively affect streamflow. While groundwater flow parameters significantly influence groundwater heads.

Snow parameters include Melt factor for snow on June 21, Snowmelt base temperature, streambed thickness, Snowpack temperature lag factor, and curve number for surface runoff processes. Groundwater flow parameters hydraulic conductivity and specific yield.

4) Cache River (extensive groundwater pumping for irrigation): soil water related parameters significantly affect streamflow including Soil evaporation compensation factor and percolation coefficient. Meanwhile, groundwater

flow and surface runoff have parameters a relatively less influence on stream discharge. For groundwater head, soil water related parameters pointedly affect streamflow comprising available water capacity and Soil evaporation compensation factor.



5) The iES method represents prior parameter ensembles (uncalibrated Monte Carlo results) with wide uncertainty band, and the posterior ensemble effectively reduces the uncertainty band. This technique can give best estimation parameter ranges, water balance components, and simulated streamflow and groundwater heads.

While these SA-UA-PO methods have been demonstrated here for the SWAT+ *gwflow* model, they can be applied generally to other coupled surface-subsurface models, or even stand-alone watershed models such as SWAT or SWAT+.

**Acknowledgements**

The United States Department of Agriculture–Agricultural Research Service provided funding for this study through Cooperative Agreements 59-3098-8-002 and 59-3098-2-001.USDA is an Equal Opportunity Employer and Provider.

**Author contributions:**

The research was designed by S.A., R.B., and J.W., with model coding performed by R.B., J.W., and J.A. S.A., R.B., J.W., M.W., J.A., N.C., and J.G. conducted the research, and R.B., S.A., J.G., and N.C. analyzed the data. The paper was written by S.A., R.B., J.W, and M.W.

**Conflicts of Interest**

The authors declare no conflict of interest.

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



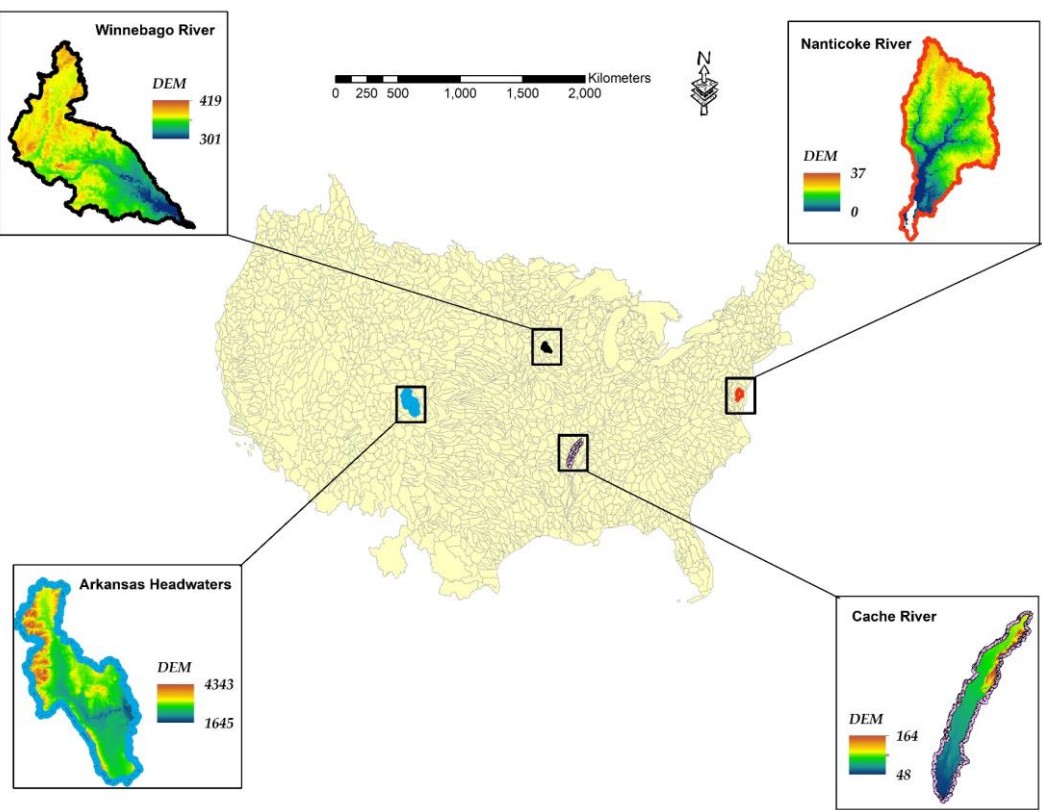

**Figure 1: Geographical locations, and digital elevation model of the four study watersheds. Arkansas Headwaters River; Winnebago River; Nanticoke River; Cache River.**





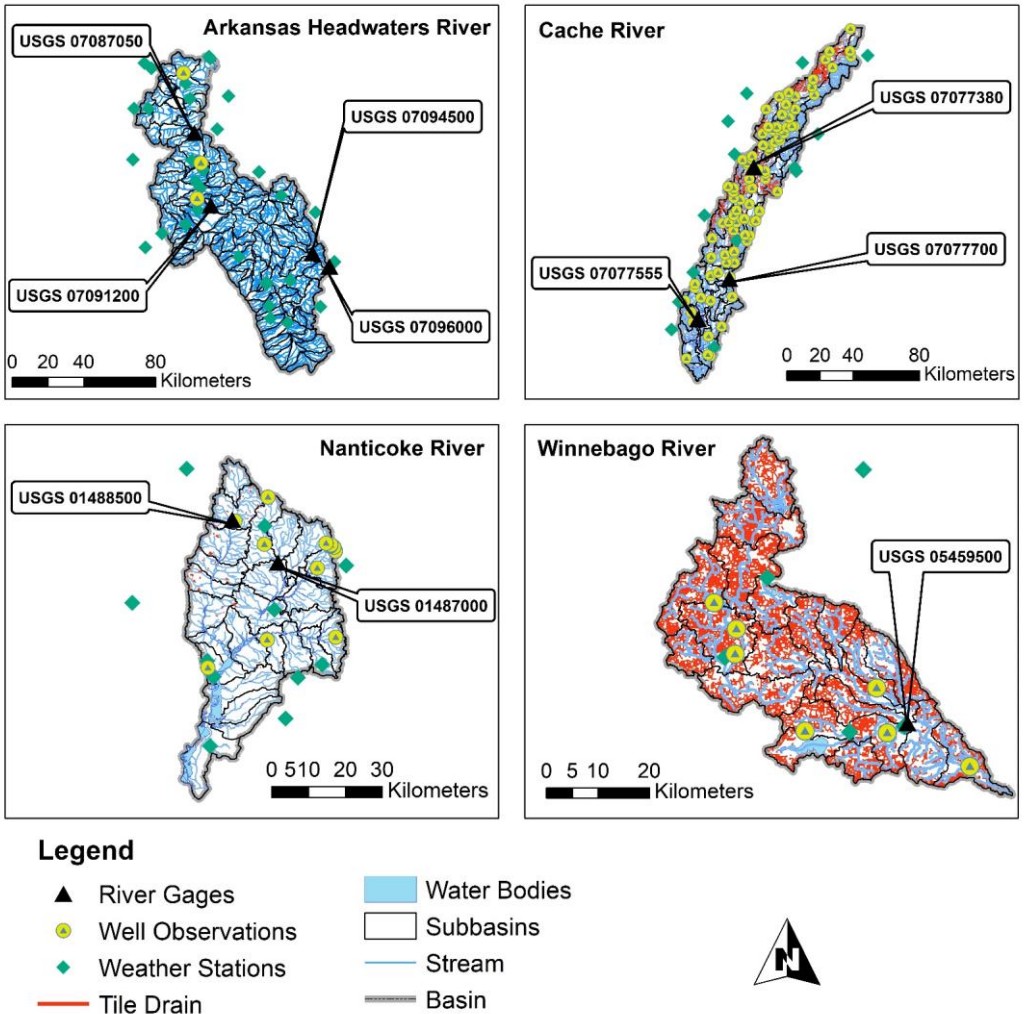

**Figure 2: Detailed maps of the study watersheds, revealing the location of water bodies, streams, USGS monitoring wells, weather stations, subbasin boundaries, tile drains, and river gages stations.**





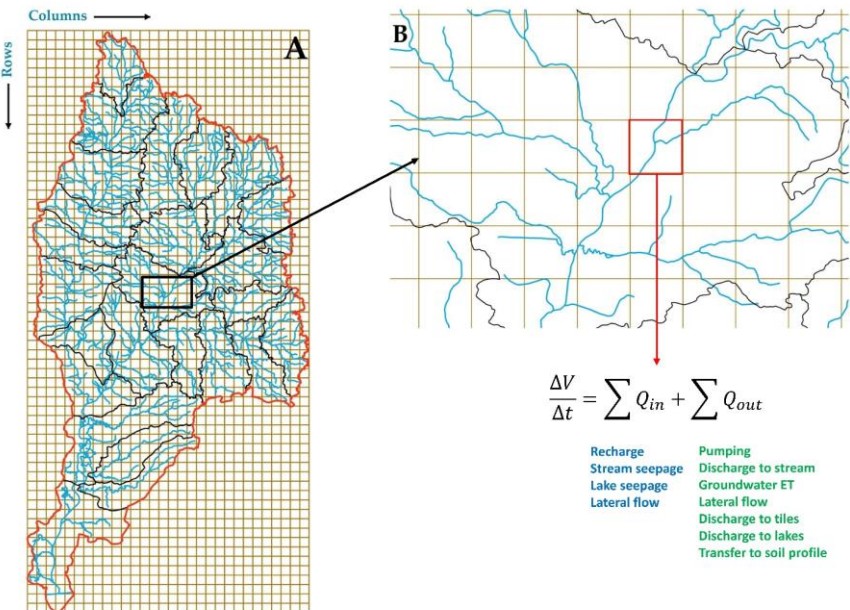

**Figure 3: Geographical layout and computation method of the *gwflow* module, presenting (A) grid cells, watershed boundary (red line), stream channels (blue lines), and subbasins (black lines) for the Nanticoke watershed; and (B) Zoomed-in of channels and grid, demonstrating the water balance computations for each cell.**

585

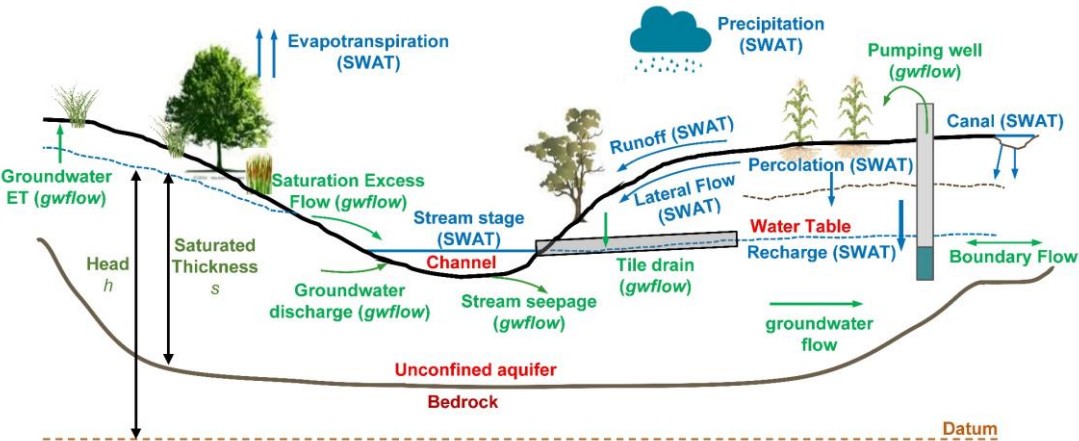

**Figure 4: Schematics representation of the hydrologic processes in a typical watershed stream-aquifer system showing main hydrologic elements and hydrologic processes for SWAT+ and *gwflow*. Blue arrows outline fluxes that are calculated by SWAT+, green arrows for flow processes that are computed by *gwflow*.**

590



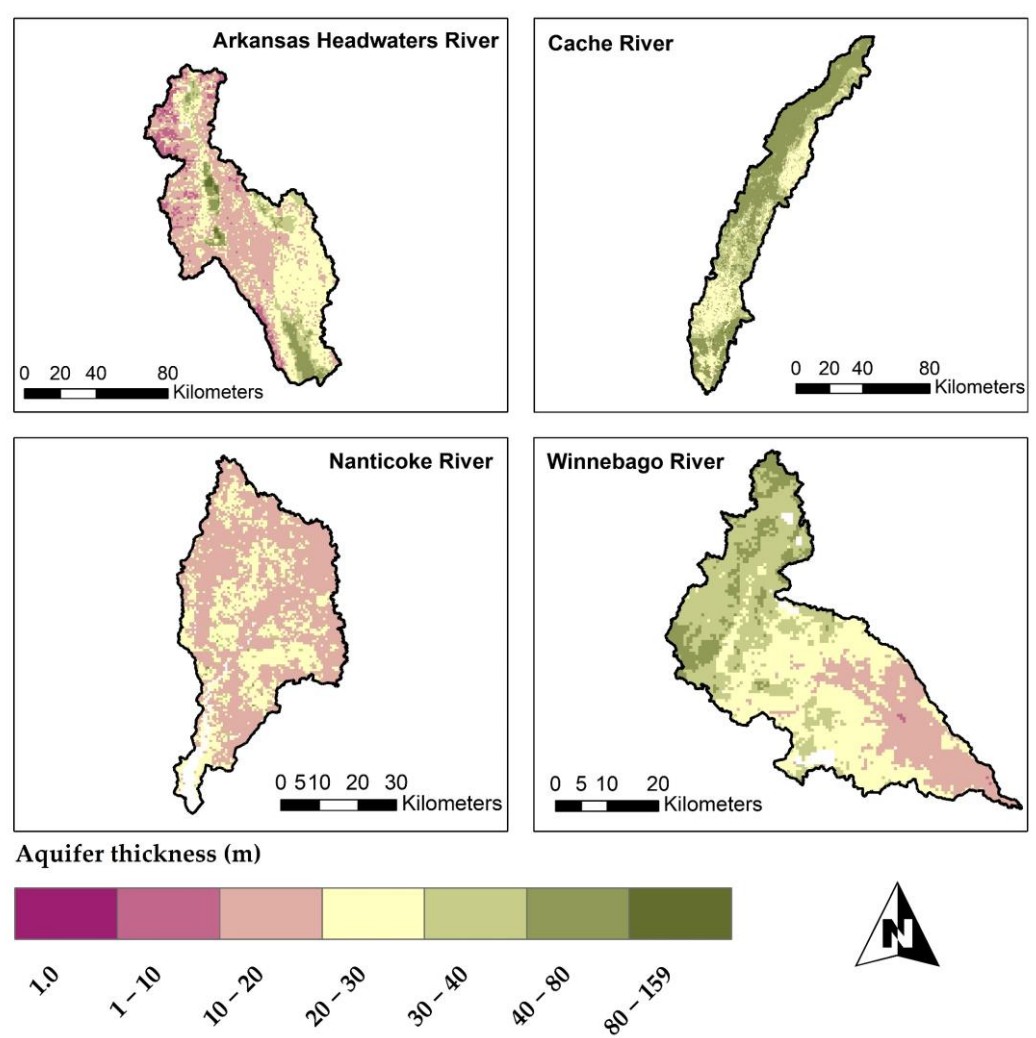

**Figure 5: Schematics representation aquifer thickness (m) maps for the four study watersheds of each grid cell.**





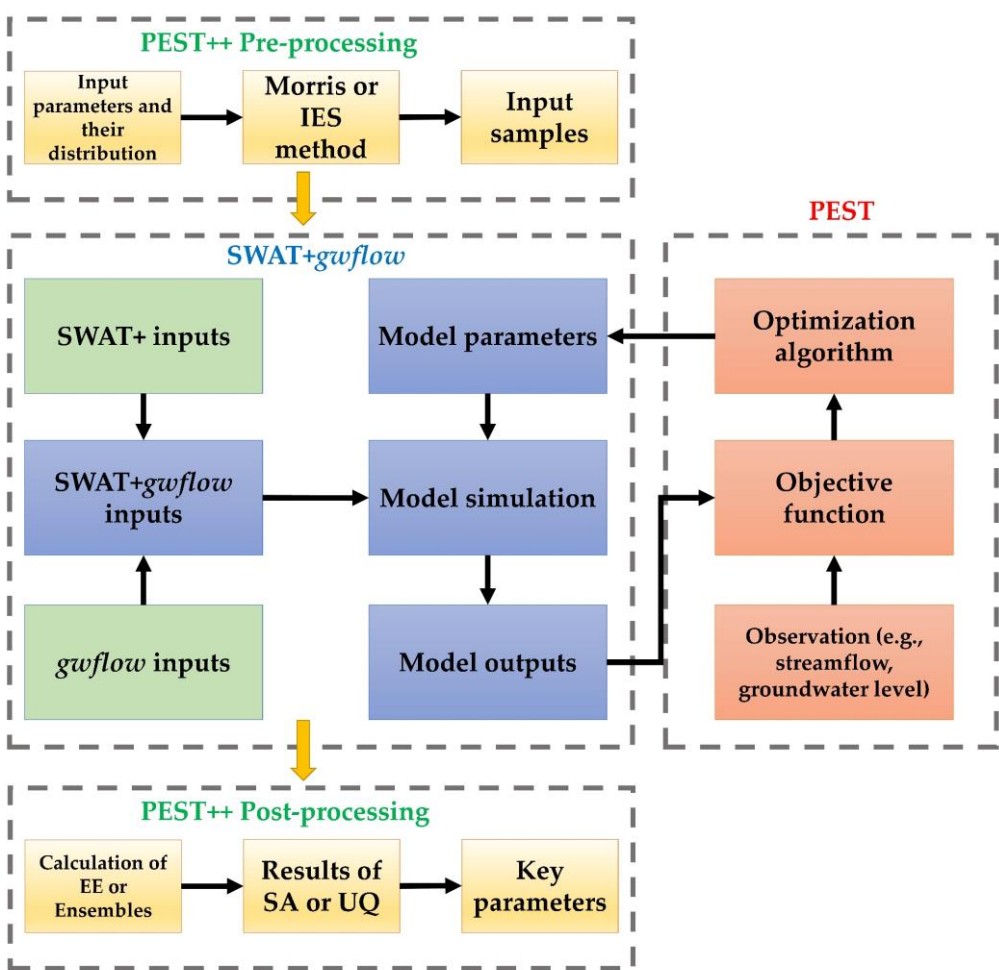

**Figure 6: Schematic of PEST automatic calibration, sensitivity analysis, and uncertainty analysis (iES) applied to the SWAT+*gwflow* models.**





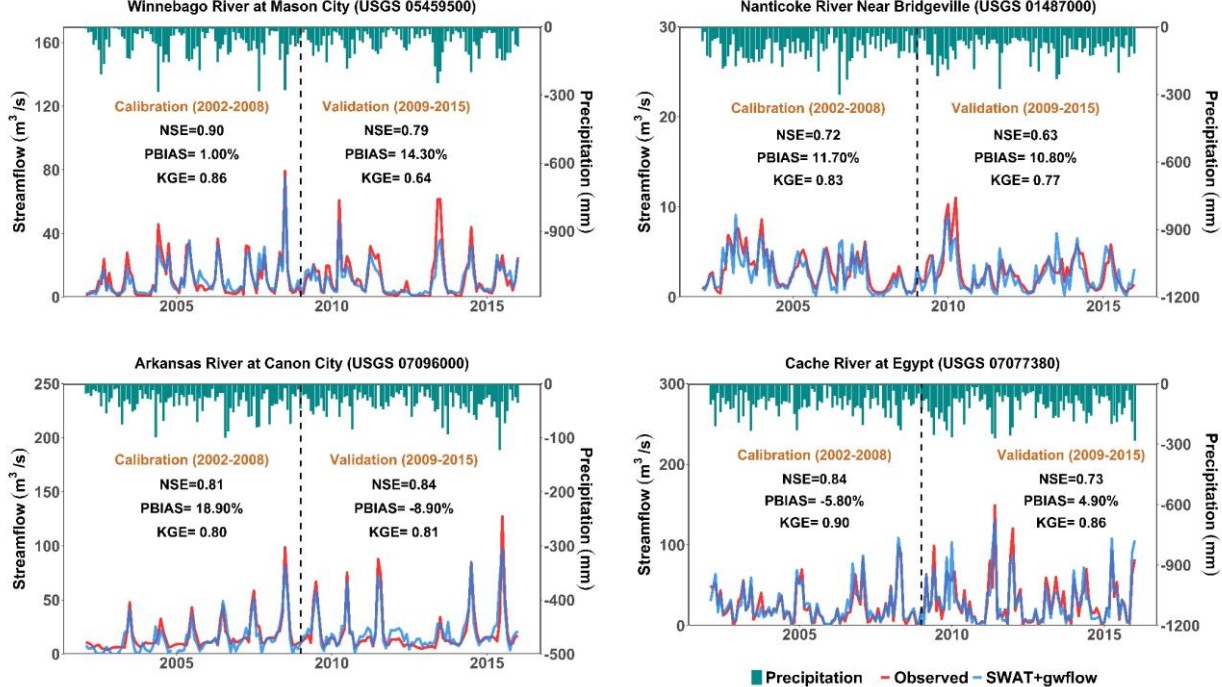

**Figure 7: Measured and simulated monthly streamflow for SWAT+*gwflow* models for four selected river gage stations within the four study watersheds. Statistical model performances (NSE, PBIAS, KGE) are presented for each gage location.**



**Figure 8: Monthly surface water fluxes (mm) [left column], and groundwater fluxes (mm) [right column] for the simulation period of (2002–2015) for the four study watersheds.**





**Figure 9:** Maps showing statistical model performance based on mean absolute error (MAE) (m) for groundwater level for the simulation period of (2000–2015) in the study watersheds.





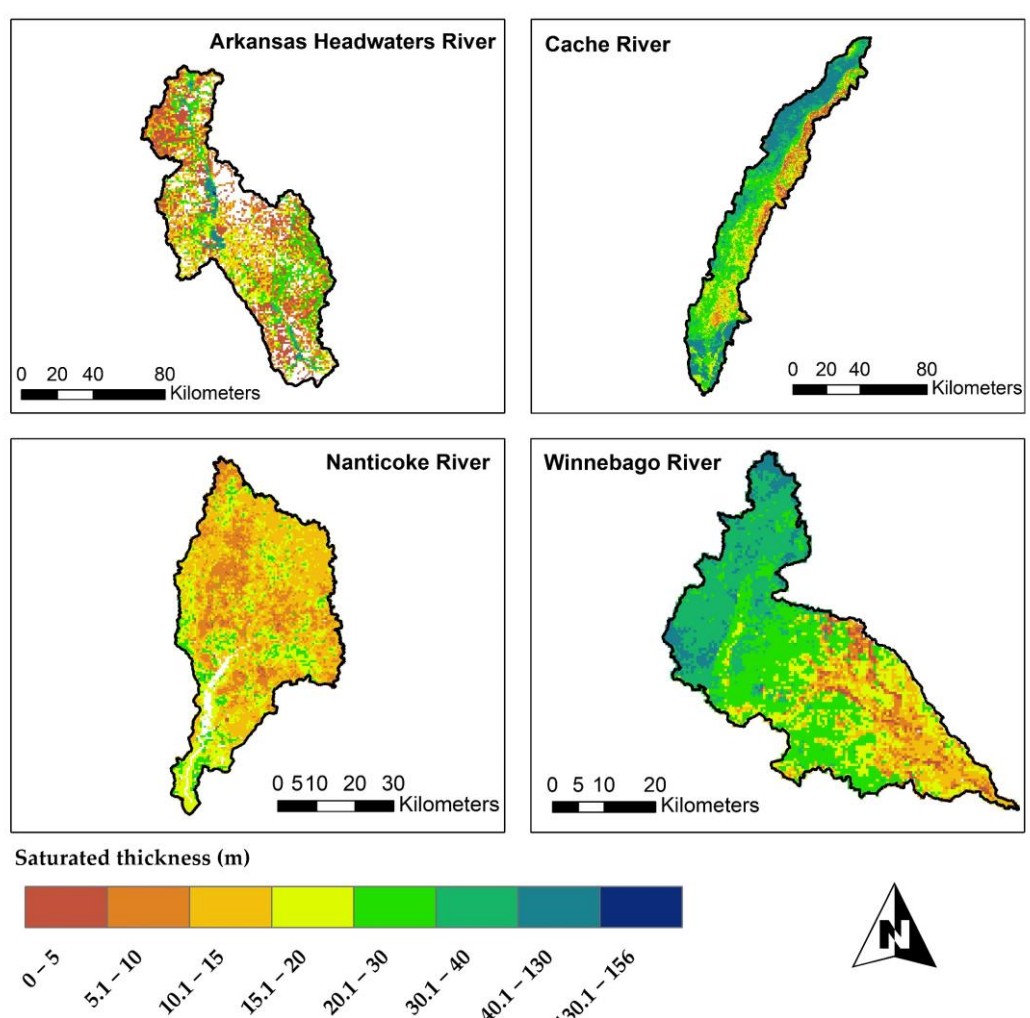

**Figure 10: Maps of saturated thickness (m) in the four study watersheds.**





**Figure 11: Maps of average annual recharge flow (m3/day) for the period of (2000–2015) for each of the study watersheds for each**
615 **grid.**

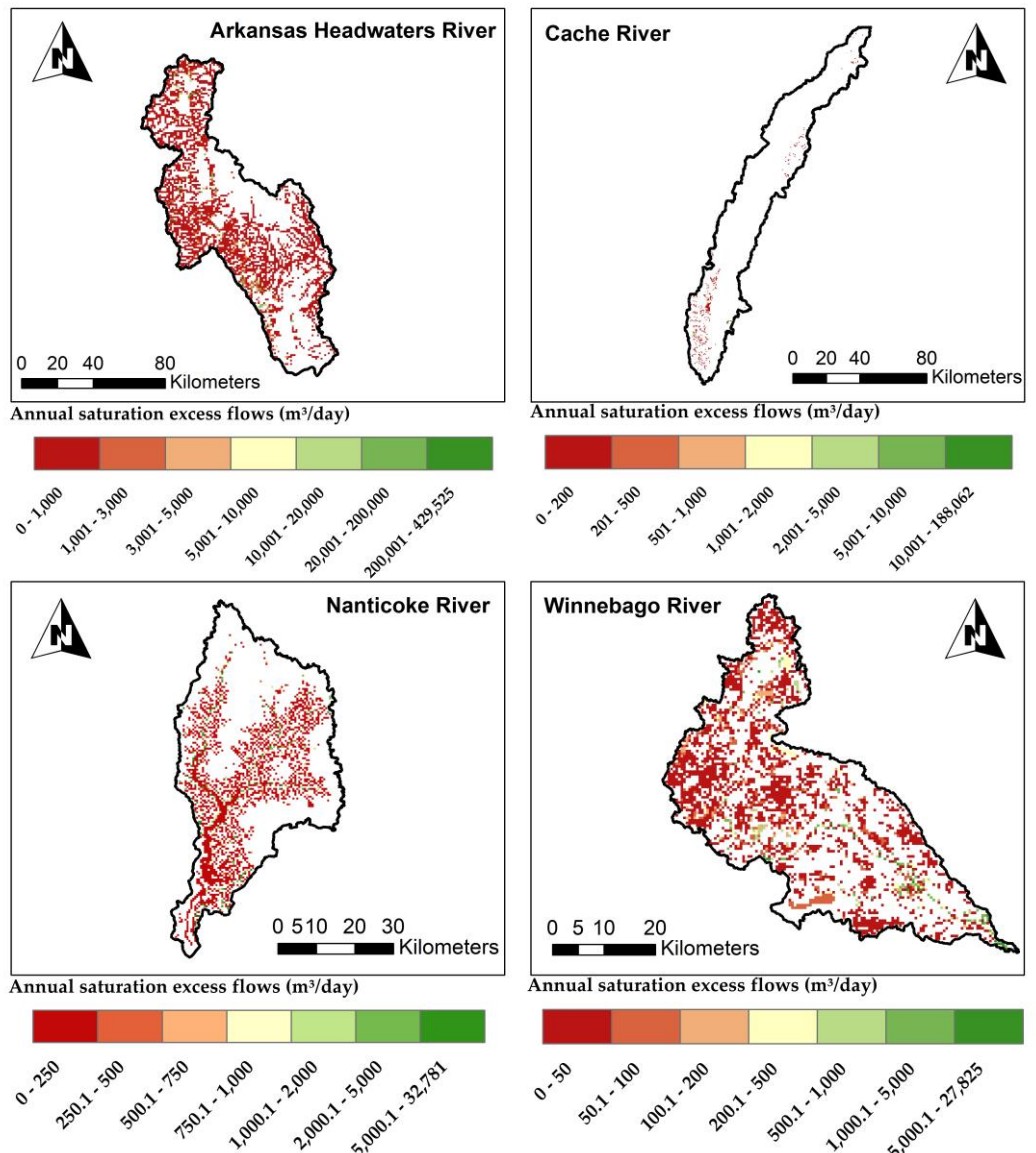

**Figure 12: Maps of average annual saturation excess flow (m3/day) for the period of (2000–2015) in each of the four study watersheds for each grid.**





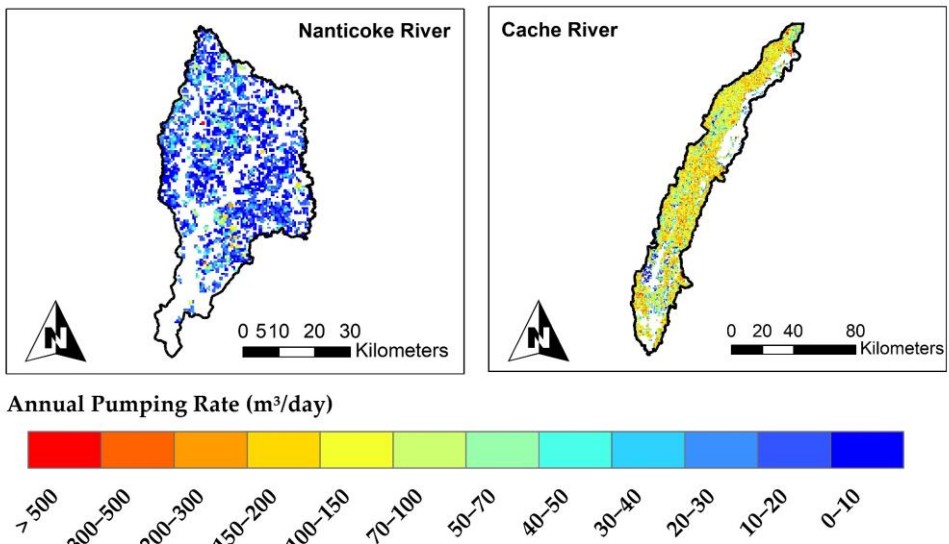

**Figure 13: Maps of average annual groundwater pumping for irrigation (m3/day) in the Cache, and Nanticoke watersheds for each grid cell.**

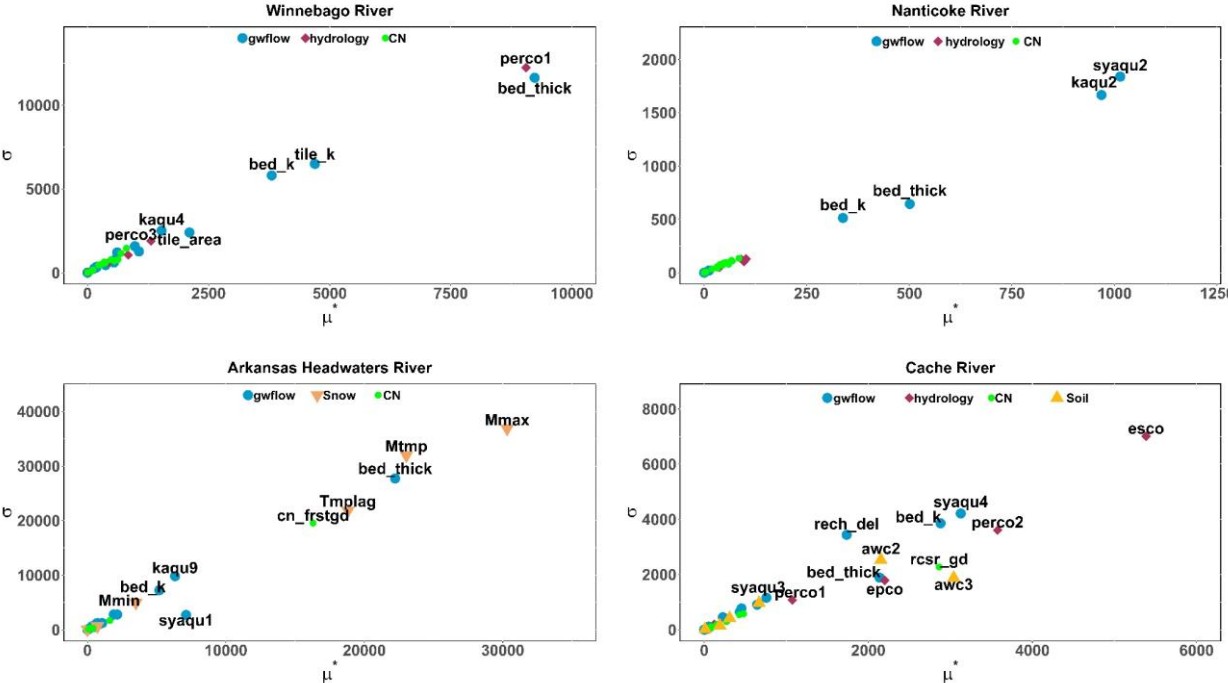

**Figure 14: Parameters sensitivity analysis based on the Morris screening method for minimizing streamflow. Only the most sensitive parameters are labelled. σ reveals the degree of nonlinearity or factor interaction, and μ* is the sensitivity measure.**

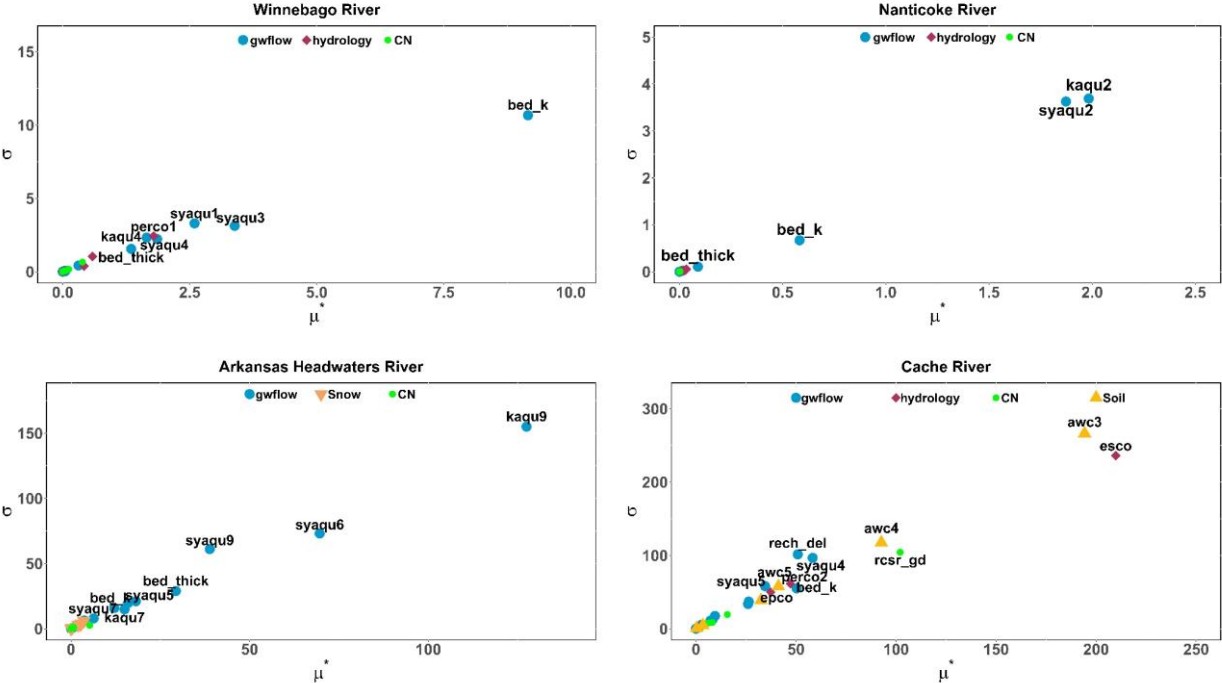

625

**Figure 15: Parameters sensitivity analysis based on the Morris screening method for minimizing groundwater level. Only the most sensitive parameters are labelled. σ reveals the degree of nonlinearity or factor interaction, and μ* is the sensitivity measure.**





**Figure 16: Estimated sensitivity that changes with time for the streamflow for the key influential parameters of the four study watersheds. The blue lines represent *gwflow* parameters, maroon lines for hydrology parameters, and orange lines represent the snow parameters.**

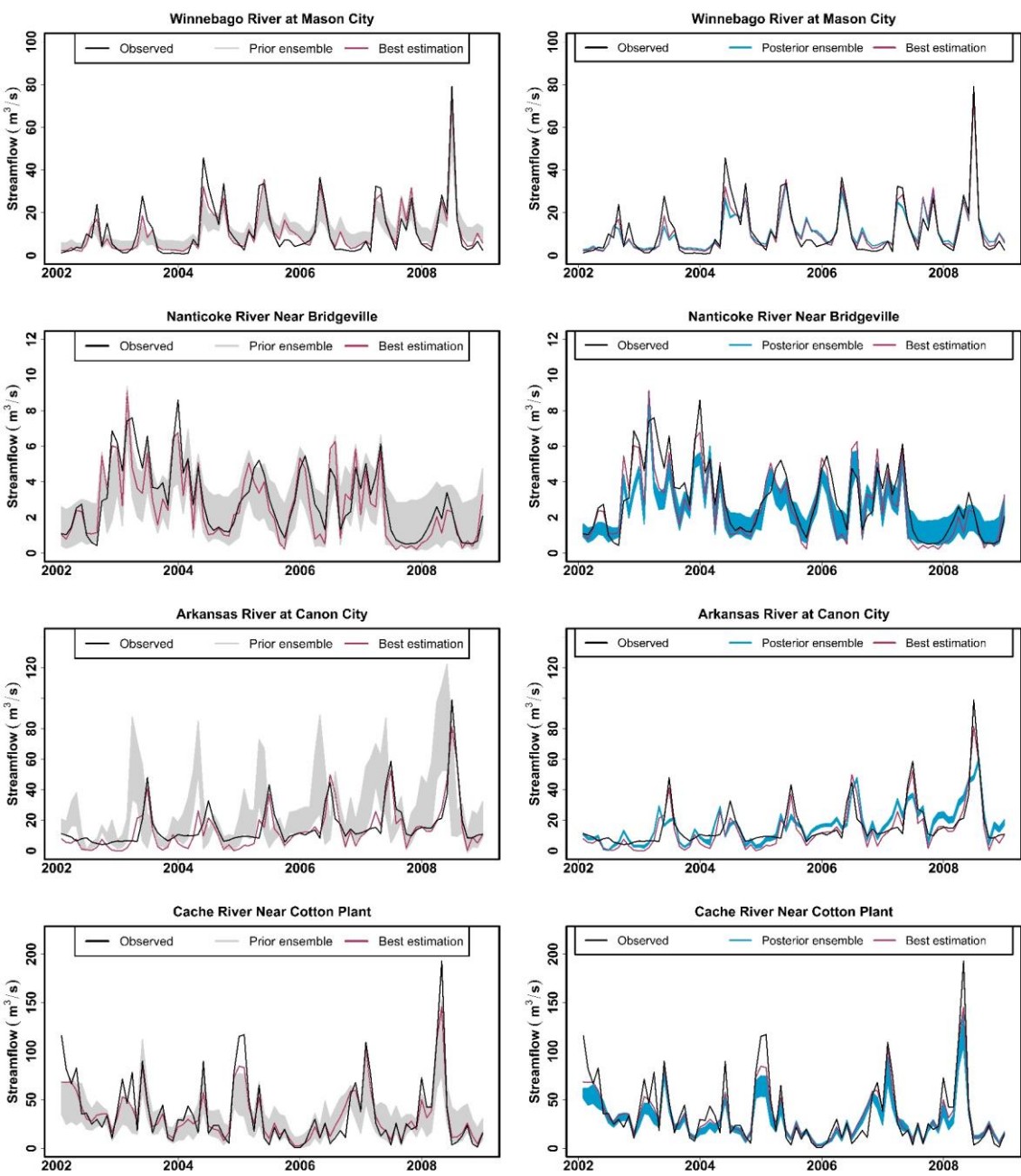

**Figure 17: Prior (left column) and posterior (right column) prediction uncertainty bounds for streamflow estimation for SWAT+*gwflow* for four study watersheds.**





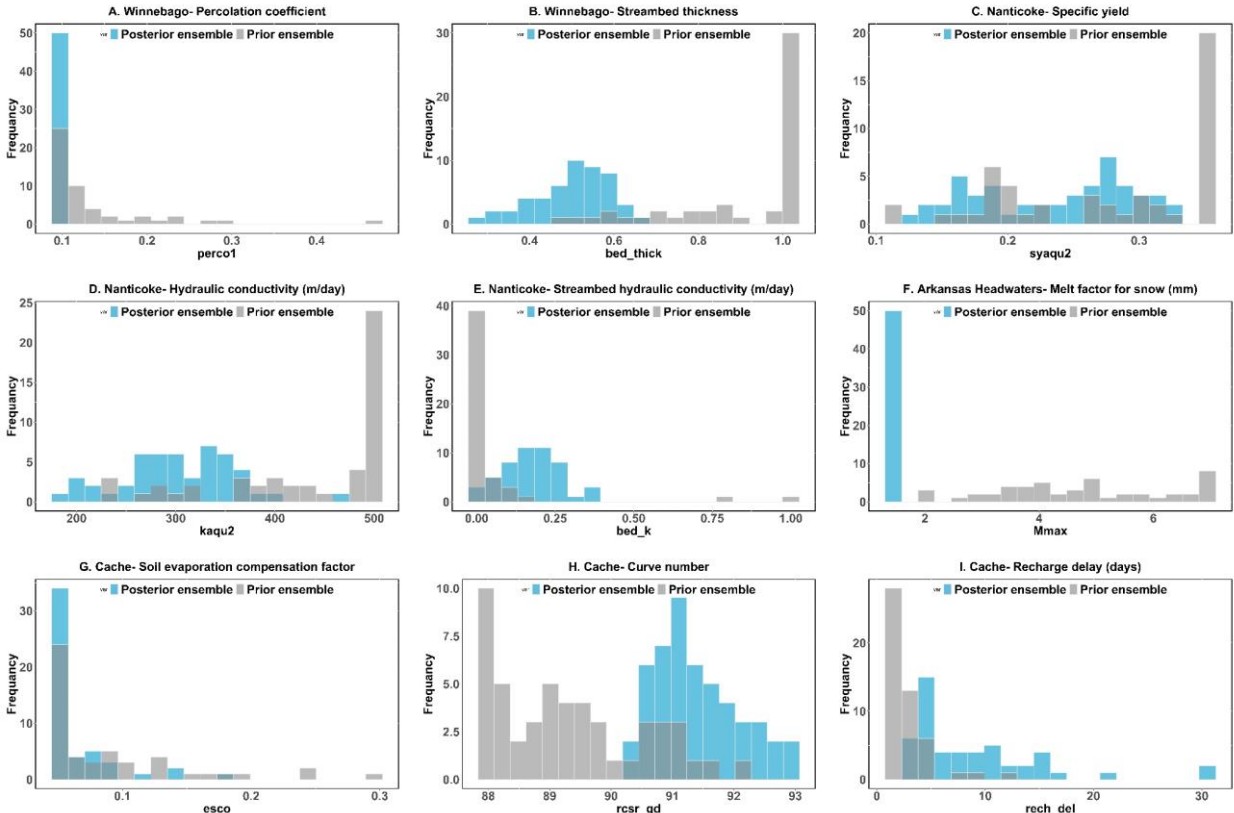

635

**Figure 18: Histogram for prior and posterior for significant parameters for four study watersheds.**





**Figure 19: Histogram for prior and posterior average annual water balance component (significant components) for four study watersheds.**





**Table 1.** Key features for the four study basins.

| Watershed | State | HUC2 Region | HUC8 | # Channels | # HRU | Annual Precip. (*mm*) | Area (*km²*) | Rows | Cols | Cell size (m) |
|---|---|---|---|---|---|---|---|---|---|---|
| Winnebago | IA, MN | Upper Mississippi | 07080203 | 437 | 4358 | 880 | 1787 | 140 | 139 | 500 |
| Nanticoke | DE, MD | Mid Atlantic | 02080109 | 1069 | 5519 | 1180 | 2142 | 186 | 90 | 500 |
| Arkansas Headwaters | CO | Arkansas-White-Red | 11020001 | 2230 | 2986 | 425 | 7940 | 180 | 110 | 1000 |
| Cache | AR, MO | Lower Mississippi | 08020302 | 2941 | 17143 | 1287 | 5198 | 428 | 222 | 500 |

**Table 2.** Datasets utilized to create the *gwflow* inputs and the base SWAT+ models (Bailey et al., 2023)

| | Dataset | Resolution (m) | Source |
|---|---|---|---|
| **SWAT+ model** | Land use, Land cover | 30 | U.S. Geological Survey, National Land Cover Data |
| | Field boundaries | | Yan and Roy (2016) |
| | Topographic slope map | 10 | USGS National Elevation Dataset (Gesch et al., 2018) |
| | Weather | | Global historical climatology network; PRISM |
| | Soil boundaries and properties | 10 | Soil Survey Staff (2014) |
| | Stream segments (NHD+) | | Moore and Dewald (2016) |
| | Crop rotation | | USDA–NASS, CDL |
| | Lakes and reservoirs | | Moore and Dewald (2016) |
| | Water use | | Dieter et al. (2018) |
| | Discharge from facilities | | Skinner and Maupin (2019) |
| **gwflow module** | Groundwater head | Vector Points | Bailey and Alderfer (2022) |
| | Aquifer thickness | 250 | Shangguan et al. (2017) |
| | Tile drainage | 30 | Valayamkunnath et al. (2020) |
| | Geologic units | Vector Polygons | Horton et al. (2017) |



**Table 3.** Description of hydrological fluxes of **23** selected parameters for the SWAT+ *gwflow* model.

| Parameters | Description of Parameter | Controlled Hydrologic Processes |
|---|---|---|
| CN2 | SCS runoff curve number | Surface runoff processes (cn) |
| ESCO | Soil evaporation compensation factor | Potential and actual |
| EPCO | Plant uptake compensation factor | evapotranspiration processes (hydro) |
| rech_del | Recharge delay (days) | |
| Kaqu # | Aquifer hydraulic conductivity for a specific zone (m/day) for $i^{th}$ zone | |
| Syaqu # | Aquifer specific yield for a specific zone for $i^{th}$ zone | |
| bed_k | Streambed hydraulic conductivity (m/day) | |
| bed_thick | Streambed thickness (m) | Groundwater flow processes (gwflow) |
| bed_depth | River depth (m) | |
| tile_depth | Depth of tiles below ground surface (m) | |
| tile_area | Area of groundwater inflow (m²) to tile | |
| tile_k | Hydraulic conductivity of the drain perimeter (m/day) | |
| Ftmp | Snowfall temperature (°C) | |
| Snowd | Minimum snow water content (mm $H_2O$) | |
| Mmin | Melt factor for snow on December 21 (mm $H_2O$/°C– day) | |
| Mmax | Melt factor for snow on June 21 (mm $H_2O$/°C– day) | Snow processes (sno) |
| Mtmp | Snowmelt base temperature (°C) | |
| Tmplag | Snowpack temperature lag factor | |
| COV50 | Fraction of COVMX | |
| SOL_BD # | Moist bulk density (g/cm³ or Mg/m³) for $i^{th}$ layer | |
| SOL_AWC # | Available water capacity of the soil layer (mm $H_2O$/mm soil) for $i^{th}$ layer | Soil water processes (sol) |
| Perco | Percolation coefficient | |
| SOL_K # | Saturated hydraulic conductivity (mm/h) for $i^{th}$ layer | |








**Table 4.** Selected parameters and ranges for the sensitivity and uncertainty analysis for the SWAT+ *gwflow* model.

| Watershed | Parameter | No. of classes | Hydrologic process | Parameter range |
|---|---|---|---|---|
| **Nanticoke River** | rech_del | – | gwflow | 1 to 30 |
| | Kaqu # | 2 | gwflow | –80% to +100% (relative) |
| | Syaqu # | 2 | gwflow | 0.05 to 0.35 |
| | bed_k | – | gwflow | 0.0001 to 1 |
| | bed_thick | – | gwflow | 0.2 to 1 |
| | bed_depth | – | gwflow | –80% to +20% (relative) |
| | CN2 | 4 | cn | 0 to +30% (relative) |
| | esco | – | hydro | 0 to 1 |
| | epco | – | hydro | 0 to 1 |
| | perco | 2 | hydro | 0 to 1 |
| **Winnebago River** | rech_del | – | gwflow | 1 to 30 |
| | Kaqu # | 4 | gwflow | –90% to +100% (relative) |
| | Syaqu # | 4 | gwflow | 0.05 to 0.35 |
| | bed_k | – | gwflow | 0.0001 to 1 |
| | bed_thick | – | gwflow | 0.2 to 1 |
| | bed_depth | – | gwflow | –80% to +20% (relative) |
| | tile_depth | – | gwflow | 1 to 2 |
| | tile_area | – | gwflow | 10 to 100 |
| | tile_k | – | gwflow | 0.5 to 15 |
| | CN2 | 4 | cn | –12 to +12% (relative) |
| | esco | – | hydro | 0 to 1 |
| | epco | – | hydro | 0 to 1 |
| | perco | 3 | hydro | 0 to 1 |
| **Cache River** | rech_del | – | gwflow | 1 to 30 |
| | Kaqu # | 5 | gwflow | –80% to +100% (relative) |
| | Syaqu # | 5 | gwflow | 0.05 to 0.35 |
| | bed_k | – | gwflow | 0.0001 to 1 |
| | bed_thick | – | gwflow | 0.2 to 1 |
| | bed_depth | – | gwflow | –80% to +20% (relative) |
| | tile_depth | – | gwflow | 1 to 2 |
| | tile_area | – | gwflow | 10 to 60 |
| | tile_k | – | gwflow | 0.5 to 10 |
| | CN2 | 3 | cn | –7 to +33% (relative) |
| | esco | – | hydro | 0 to 1 |
| | epco | – | hydro | 0 to 1 |
| | perco | 4 | hydro | 0 to 1 |
| | awc | 6 | sol | 0 to 1 |
| **Arkansas Headwaters River** | Ftmp | – | sno | 0 to 5 |
| | Mtmp | – | sno | 0 to 5 |
| | Mmax | – | sno | 1.4 to 6.9 |
| | Mmin | – | sno | 1.4 to 6.9 |
| | Tmplag | – | sno | 0.01 to 1.01 |
| | Snowd | – | sno | 0.5 to 1 |
| | COV50 | – | sno | 0.1 to 1 |
| | CN2 | 2 | cn | –5 to +35% (relative) |
| | rech_del | – | gwflow | 1 to 30 |
| | Kaqu # | 9 | gwflow | –90% to +100% (relative) |
| | Syaqu # | 9 | gwflow | 0.05 to 0.35 |
| | bed_k | – | gwflow | 0.0001 to 1 |
| | bed_thick | – | gwflow | 0.2 to 1 |
| | bed_depth | – | gwflow | –80% to +20% (relative) |




**Table 5.** Monthly discharge statistical performance for the SWAT+ *gwflow* simulation.

| River Basin | Station | Calibration | | | | Validation | | | |
|---|---|---|---|---|---|---|---|---|---|
| | | NSE | R² | PBIAS | KGE | NSE | R² | PBIAS | KGE |
| **Nanticoke River** | USGS 01488500 | 0.79 | 0.79 | −3.30 | 0.85 | 0.81 | 0.81 | −5.40 | 0.86 |
| | USGS 01487000 | 0.72 | 0.77 | 11.70 | 0.83 | 0.63 | 0.66 | 10.80 | 0.77 |
| **Winnebago River** | USGS 05459500 | 0.90 | 0.91 | 1.00 | 0.86 | 0.79 | 0.88 | 14.30 | 0.64 |
| **Cache River** | USGS 07077380 | 0.84 | 0.85 | −5.80 | 0.90 | 0.73 | 0.75 | 4.90 | 0.86 |
| | USGS 07077700 | 0.77 | 0.81 | 13.20 | 0.76 | **Not enough observations** | | | |
| | USGS 07077555 | 0.85 | 0.91 | 6.90 | 0.73 | 0.85 | 0.92 | 14.90 | 0.70 |
| **Arkansas Headwaters River** | USGS 07087050 | 0.91 | 0.92 | −6.90 | 0.91 | 0.94 | 0.95 | −8.60 | 0.91 |
| | USGS 07091200 | 0.91 | 0.93 | 3.80 | 0.90 | 0.96 | 0.96 | 0.60 | 0.96 |
| | USGS 07094500 | 0.73 | 0.84 | 23.10 | 0.75 | 0.84 | 0.85 | 7.30 | 0.84 |
| | USGS 07096000 | 0.81 | 0.85 | 18.90 | 0.80 | 0.84 | 0.85 | −8.90 | 0.81 |


**Table 6.** Model run times for simulation period of (2000–2015) for four study areas using standalone SWAT+ and holistic SWAT+*gwflow*.

| Watershed | Base SWAT+ (Minutes: seconds) | Holistic SWAT+*gwflow* (Minutes: seconds) |
|---|---|---|
| Winnebago River | 02: 37.10 | 07: 15.12 |
| Nanticoke River | 04: 30.00 | 11: 50.88 |
| Cache River | 12: 49.57 | 34: 43.75 |
| Arkansas Headwaters River | 05: 06.01 | 13: 23.92 |






**Table 7.** Mean annual hydrologic flow processes (mm) for 4 study watersheds with main fluxes fraction.

| | Flux (mm) | Winnebago | Nanticoke | Cache | Arkansas Headwaters |
|---|---|---|---|---|---|
| **Input** | Precipitation | 880 | 1180 | 1287 | 425 |
| | Boundary Inflow | 50 | 143 | 90 | − 2.40 |
| **Watershed Output** | Surface Runoff | 103 | 256 | 421 | 51 |
| | Sat Excess Flow | 75 | 183 | 37 | 4.6 |
| | Tile flow | 62 | 2.81 | 0.07 | 0.0 |
| | Stream seepage | − 26 | − 0.38 | −1.70 | − 4 |
| | Soil Lateral Flow | 65 | 131 | 47 | 28 |
| | ET | 580 | 790 | 941 | 336 |
| **Internal Flows** | Recharge | 73 | 33 | 90 | 5.7 |
| | Pumping Irrigation | 0 | 15.5 | 141 | 0.42 |
| | GW-Lake Exchange | − 0.33 | − 0.70 | − 1.6 | − 4 |
| | Surface Water Irrigation | 0 | 1.00 | 43 | 0.14 |
| **Fractions** | Water Yield | 279 | 573 | 504 | 80 |
| | Recharge Fraction[a] | 0.08 | 0.03 | 0.07 | 0.01 |
| | Yield Fraction[b] | 0.31 | 0.48 | 0.39 | 0.19 |
| | Baseflow Fraction[c] | 0.18 | 0.32 | 0.07 | 0.00 |
| | ET Fraction[d] | 0.65 | 0.66 | 0.72 | 0.79 |

a: Recharge / Precipitation

b: Water Yield / Precipitation

c: Net groundwater inflow to streams (Stream Seepage + Sat Excess Flow) / Water Yield

d: ET / Precipitation
