# Peer review of "A Framework for Parameter Estimation, Sensitivity Analysis, and Uncertainty Analysis for Holistic Hydrologic Modeling Using SWAT+"

_Hydrology and Earth System Sciences, 2023_

## Author Comment (AC1)

**Reviewer 1**

The authors present a study in which they couple SWAT+ with *gwflow* model and examine the parameters, sensitivity, and uncertainties for four study areas in the United States. The areas differ not only in terms of climatic conditions, but also in terms of hydro(geo)logic processes such as water withdrawal, GW-SW interactions, etc.

They demonstrate how a systematic workflow using relatively recently developed tools such as iES can be used within the PEST framework to take full advantage of PEST and its utilities in calibrating groundwater and surface water models and in analyzing the uncertainties in the predictions made by these models before and after calibration.

Although I believe that more studies of this type are needed, and that uncertainties and sensitivities are still too infrequently determined systematically, I do have comments that should be included.

**Response**: Thank you! We found your comments extremely helpful. We have read your comments carefully and tried our best to address them.

1. In general, I find that the current manuscript does not discuss enough the results in a broader context (a discussion section would be extremely useful).

   a. For example, it could be discussed why bed_thickness is an important parameter in almost all study areas. Here a stronger link to existing literature should be made.

**Response**: Thank you for your comment. We have included the following discussion regarding streambed parameters, and added new references to our reference list:

Lines 372-383:

*"The strong influence of streambed parameters (streambed conductivity, streambed thickness) on system responses in each of the four study watersheds is expected due to the coupled surface/subsurface nature of the watersheds. Water exchange between channels and aquifers increases with increasing conductivity and decreasing thickness. Streambed parameters have a strong control on streamflow for each of the four watersheds, whereas they control groundwater head for only the Winnebago River watershed and the Nanticoke River watershed, due to shallow groundwater levels in relation to ground surface and channel elevation. For streamflow, control is either in the direction of channel→aquifer (seepage) or aquifer→channel (discharge). For the Cache River watershed, extensive groundwater pumping (see Figure 13) can lead to enhanced stream seepage ("streamflow depletion") which, as noted by previous studies (Fox and Durnford, 2003; Fox, 2007) can be sensitive to streambed conductivity. In general, the importance of streambed parameters such as conductivity and thickness in the modeling of surface-groundwater (SW-GW)*

*exchange fluxes have been noted extensively (Kalbus et al., 2009; Brunner et al., 2017; Partington et al., 2017), with*

*many studies aimed at quantifying these parameters spatially (e.g., Fox, 2007; Crook et al., 2008; Wojnar et al., 2013;*

*Shi and Wang, 2023)."*

**Add References**

Brunner, P., Therrien, R., Renard, P., Simmons, C., and Franssen, H.: Advances in understanding river-groundwater interactions. *Rev. Geophys.*, *55*(3), 818-854. https://doi.org/10.1002/2017RG000556, 2017.

Crook, N., Binley, A., Knight, R., Robinson, D. A., Zarnetske, J., and Haggerty, R.: Electrical resistivity imaging of the architecture of substream sediments. *Water Resour. Res.*, *44*(4). https://doi.org/10.1029/2008WR006968, 2008.

Fox, G., and Durnford, D.: Unsaturated hyporheic zone flow in stream/aquifer conjunctive systems. *Adv. Water Resour., 26*(9), 989-1000. https://doi.org/10.1016/S0309-1708(03)00087-3, 2003.

Fox, G.: Estimating streambed conductivity: guidelines for stream-aquifer analysis tests. *Trans ASABE*, *50*(1), 107-113. https://doi.org/10.13031/2013.22416, 2007.

Kalbus, E., Schmidt, C., Molson, J. W., Reinstorf, F., and Schirmer, M.: Influence of aquifer and streambed heterogeneity on the distribution of groundwater discharge. *Hydrol. Earth Syst. Sci.*, *13*(1), 69-77. https://doi.org/10.5194/hess-13-69-2009, 2009.

Partington, D., Therrien, R., Simmons, C., and Brunner, P.: Blueprint for a coupled model of sedimentology, hydrology, and hydrogeology in streambeds. *Rev. Geophys.*, *55*(2), 287-309. https://doi.org/10.1002/2016RG000530, 2017.

Shi, W., and Wang, Q.: An Analytical Model of Multi-layered Heat Transport to Estimate Vertical Streambed Fluxes and Sediment Thermal Properties. *J. Hydrol.*, 129963. https://doi.org/10.1016/j.jhydrol.2023.129963, 2023.

Wojnar, A., Mutiti, S., and Levy, J.: Assessment of geophysical surveys as a tool to estimate riverbed hydraulic conductivity. *J. Hydrol.*, *482*, 40-56. https://doi.org/10.1016/j.jhydrol.2012.12.018, 2013.

    b.   Also, a comparison/discussion to integrated models could be made (such as, HGS, Parflow, etc.) and

**Response**: Thanks for your comment. We have included the following discussion:

*"These fast computation times greatly facilitate calibration, sensitivity analysis, and uncertainty analysis for our regional-scale hydrologic models. Other physically based holistic hydrologic models could be used (e.g., HydroGeoSphere, Parflow, mHM), but required heterogeneous parameters and long computation times are often prohibitive for the hundreds and thousands of simulations runs that are required for the sensitivity analysis and uncertainty analysis conducted in this study."*

    c.   the assumptions of the coupling SWAT+*gwflow* for the groundwater-surface water interaction could be discussed.

**Response**: Yes, we agree. We have included several features and limitations of the modeling set-up:

*"As with the initial set-up of these models, the following features, and limitations of the SWAT+gwflow modelling framework, as used in this study, should be noted:*

1. *The gwflow module only considers a single-layer heterogeneous unconfined aquifer, in connection with the network of fields, channels, and reservoirs.*

2. *Recharge from cultivated fields to the unconfined aquifer is explicitly simulated; however, recharge from non-field HRUs is not spatially explicit, as the delineation of these HRUs is not provided in the NAM. Therefore, recharge for non-field areas is calculated using the average recharge rate for the 12-digit catchment.*

3. *The gwflow module does include an option to move water from the aquifer to the soil profile of the HRU if the water table rises above the base of the soil profile; using this process, shallow groundwater can be used as crop ET or discharged to nearby channels via soil lateral flow. However, due to the lack of spatial representation of non-field HRUs in the NAM, the groundwater→soil option is not possible. Therefore, shallow groundwater is allowed to rise to the ground surface and, if groundwater head increases above the ground surface, the volume of water above the ground is routed as saturation excess flow to the nearest channel. We acknowledge this simplification but believe the methods to be adequate in regional-scale applications."*

4. *Groundwater fluxes along the boundary of the watershed are simulated using a boundary condition approach: the groundwater head in cells along the watershed boundary is assumed to be fixed at the initial value at the*

*beginning of the simulation. If the cell head value is higher than adjacent head values, then groundwater*

*inflow is simulated; if lower, than groundwater outflow is simulated. These fluxes are not calibrated per se,*

*but indirectly as groundwater head values within the watershed are targets in model calibration."*

2. Model simplifications can be extremely important (even if difficult to quantify) and should be discussed more. Otherwise, it seems that with the presented approach all uncertainties are known very precisely, which is not the case.

**Response**: Thank you for the comment. As noted in a previous comment, we have outlined the key simplifications on **lines 186-204**. We also added another section on **lines 334-338**.

**Lines 334-338**:

*"Within the SWAT+gwflow framework, stream seepage and groundwater saturation excess runoff constitute groundwater-stream interaction. Throughout the stream system, seepage to the aquifer occurs, with the highest rates typically found along the major rivers because of the large head difference between the stream and the surrounding water table at those locations. High values of saturation excess runoff can be found in the vicinity of rivers and streams in areas of shallow groundwater levels."*

3. Comment 3

   a. Simplifications and limitations in the calibration should also be discussed more intensively.

**Response**: Thank you for the comment. As noted in a previous comment, we have outlined the key simplifications on **lines 256-259** and **lines 277-278.**

**Lines 256-259**:

*"PEST is a powerful inverse modeling tool that can handle many parameters, needs linearity and a stable model, and requires several methods for parameter adjustment. However, determining the minimum of the objective function is restricted if there is a large amount of data error, the model does not represent the data well, and there is a high degree of correlation between the parameters."*

*"In general, the data assimilation approach assumes prior and posterior multivariate Gaussian parameter distributions."*

    b.   Calibration against GW annual average values are certainly not optimal. Especially since it has often been shown that a calibration against groundwater levels alone is often not sufficient (see for example Schilling et al 2019).

**Response**: Yes, we agree. For this reason, we have calibrated and tested against joint streamflow and groundwater head. In addition, we believe that the annual average groundwater head values capture the main groundwater fluctuations in each watershed, and therefore are adequate to estimate aquifer hydraulic conductivity and specific yield and verify that the correct groundwater storage changes are being captured by the model.

    4.   Comment 4

    a.   Also, I am not sure how the calibration has dealt with parameters that are correlated. Is this corrected by the ensemble approach or do the dependencies still exist? Some more information would be useful.

**Response**: Thank you for the comment. We have included the following discussion**.**

**Lines 406-416**

*"In general, ensemble-based data assimilation naturally accommodates parameter correlation, both in the prior parameter distribution (as expressed in the prior parameter covariance matrix), as well as correlations between parameters that give correlated responses to historic observations. The former is addressed simply by providing the requisite covariance matrix or by generating a prior parameter ensemble that is imbued with appropriate parameter relations. The latter correlations, those typically referred as non-uniqueness, are handled algorithmically through the truncated SVD solution to as a mechanism to stable is the inverse problem, as well as implicitly through the use of an*

*ensemble that is naturally rank deficient (in that it does not fully occupy the range space of the parameter space). The rank deficient ensemble used to approximate the Jacobian matrix only occupies the dominant singular components of the full Jacobian – these dominate singular components are a subspace the includes the parameter combinations that represent parameters that are nonunique with respect to the historical observations. It is worth noting one of the strengths of an ensemble-based approach to history matching is that the posterior parameter spans this non-uniqueness."*

 b.    Also I don't quite understand how the flux from adjacent aquifer was determined. Is this flux also calibrated?

**Response**: The groundwater flux (across the watershed boundary) from adjacent watersheds is calculated using a boundary condition method: the groundwater head in cells along the watershed boundary is assumed to be fixed at the initial value at the beginning of the simulation. If the cell head value is higher than adjacent head values, then groundwater inflow is simulated; if lower, than groundwater outflow is simulated. These fluxes are not calibrated per se, but indirectly as groundwater head values within the watershed are targets in model calibration. This information is now included in the "features" section, on **lines 186-204**.

 c.    From Table 4 it does not seem so but since I am not sure of all the abbreviations, it might be good to at least include in the SI the full name with a small description of the parameters from Table 4.

**Response**: Thank you for pointing this out. The full name of parameters and the controlled hydrologic processes are now all in Table 3.

Other minor comments.

Line 60ff: How is this different from the Null-Space Monte Carlo approach (e.g., Alberti et al.2018, Herckenrath et al., 2011, Moeck et al., 2020) One-two sentences would certainly be useful.

**Response**: Yes, we agree. We have included the following section.

**Lines 69-78**

*"Null-Space Monte Carlo approach (NSMC) is not dissimilar to the iES approach in their goals: to represent posterior parameter uncertainty, especially as it relates to null–space parameters and parameter components (i.e., nonunique parameters). However, NSMC uses a full rank Jacobian filled using finite difference perturbations, linearized at the final calibration parameter set to project a prior parameter ensemble, realization by realization, toward being "calibrated" under the assumption of linearity. In contrast, the iES approach propagates the prior parameter ensemble directly during history matching and avoids filling a full–rank Jacobian, and instead uses an ensemble–approximation the Jacobian, an approximation that is more regional or even global, compared to the linearized local Jacobian used in NSMC. Because of this, ensemble methods can, in general, cope with higher levels of nonlinearity in the relation between parameters and observations and can also scale to much larger numbers of parameters (since the relation between number of parameters and number of model runs is removed). "*

Line 95ff: The calibration/validation selection is relatively classic (which is ok) but perhaps more information could be gleaned from the data when "extremes" are considered. This is certainly something that could be addressed in a discussion.

**Response**:

In general, most models of natural system will struggle to predict the response of previously unseen extreme events, due to step changes in the active physical processes that in the category of "unknown unknowns". This is true of the analyses presented herein. However, for historic extreme events captured observations, one could tune the data assimilation process to place additional importance on simulating the response to the extreme events, so that at least the model has been tuned to better replicate previous extreme events. This increased importance could be easily implemented by varying the weights supplied for historic responses to extreme events. We can address this in future studies.

Line 191ff: Monthly flow data is used, but the time series can be decomposed to get different components such as monthly or event quantities; base and fast flow, flow duration statistics, etc. to use in a multi-component objective function. If done properly, the processing behind each component should distill information of a different type from the measurement data set. Each of these components should therefore inform different parameters or groups of parameters. This can be done for surface water but also for groundwater data (Heudorfer et al., 2019).

**Response**: Thank you for the comment. This is a great point, and we are planning to do a similar decomposition for a future study on parameter estimation for rapid, river basin scale modeling. For this current study, however, we would prefer to use our current methods of analysis.

Figure 2: How are the data be used between the different weather stations (simple interpolation)?

**Response**: This model uses only a single weather gage data from nearest station to the centroid of each subbasin, which is then corrected using elevation band technique. This approach may result in inaccurate representation subbasin climate variable (e.g., precipitation, temperature, etc.), especially in catchments with complex topography.

Figure 14: Even if the general trend is clear and also described how the calculation takes place, I have my difficulties with the values on the x and y axes. For example, the values of mu vary from 0 to 1250 or 30000 depending on the study area. Does this say anything about the significance/sensitivity? Some more context would be extremely useful in the text.

**Response**: The quantities shown from the method of Morris analysis are the so-called "elementary effects" and are not related to the scale and magnitude of the input or output quantities. It's the relative relation between points (and the quadrants where the point lies) that are important for interpretation. A description is added in the caption of Figure 14 and Figure 15.

Figure 16: Seasonality in some parameters should be discussed in the text and also why there is a trend in bed_thick for Winnebago and bed_k for Cache.

**Response**: Thank you for your comment. Based on this comment, we reassessed the results for the Winnebago River watershed model. In doing so, we found that stream seepage was continually increasing through time, which was not realistic and which we found was an artifact of having a streambed conductivity that was far too high. Therefore, we re-ran the sensitivity analysis for this model, with a lower initial streambed conductivity value and a more realistic range of values. We have updated Figure 16 to reflect these new results. A description is added in lines 365-371.

**Lines 365-371**

*"The estimated time-varying parameter sensitivity calculated by the Morris method are represented in Fig. 16 for the most influential parameters in the four watersheds. These values are a combination of streambed parameters (bed_k), soil parameters (perc1, esco), snow parameters (Mmax, Mtmp), and aquifer parameters (syaqu2, kaqu2), depending on the watershed. The Nanticoke River model is dominated by aquifer parameters due to shallow groundwater levels and associated groundwater discharge to the stream network. The Arkansas River model is dominated by snow parameters due to mountainous terrain in the Rocky Mountains. These results indicate that these parameters have a seasonal fluctuation in their influence on streamflow, due to the seasonal fluctuations and timing of groundwater levels, snowfall, and crop growth."*

Figure 17: Even with the posterior ensemble some observed streamflow data cannot reproduced. Does that mean the model is too simple for some events?

**Response**: Thank you for your comment. We acknowledge that the model does not capture some of the streamflow temporal patterns of magnitudes. This seems normal for a hydrologic model application. Therefore: yes, the model is either too simple for some events, or the weather data input is not adequately measured and defined. However, this is always the case with hydrologic modeling.

Table 7: why is stream seepage negative?

**Response**: "Stream seepage" is listed in the "Watershed Output" section. Because it is moving water from the channel to the aquifer (and therefore the water is no longer available as a watershed output), it is given a negative sign. All other terms in this section are positive, indicating that water is leaving the watershed via the outlet.

Moreover, the recharge values seems to be very small compared to a visual (and only simple) comparison to the Reitz et al. (2017) data. It does not mean that the Reitz data are better (or the method) but please check if the data you simulated are in a plausible range.

**Response**: Thank you for the comment. Recharge fractions (recharge/precipitation) are between 1% (Arkansas River Headwaters) and 8% (Winnebago River). These values are constrained by matching both groundwater head and streamflow. We have compared (loosely) these values to regional modeling studies and, for example, the 90 mm/y for Cache is consistent with other regions within the Mississippi Alluvial Plain. Furthermore, the Reitz et al. (2017) recharge estimates are based on baseflow, which in our study is a combination of saturation excess flow and soil lateral flow, which are much higher than the recharge term. If we compare our simulated saturation excess flow + soil lateral flow, then we are much closer to the higher values reported by Reitz et al. (2017).

---

## Author Comment (AC2)

**Reviewer 2**

This paper focuses on uncertainty and sensitivity analysis techniques applied to the holistic SWAT+ model, which is coupled with a new gwflow module for physically based groundwater flow modeling. The study evaluates these techniques in four different watersheds across the United States, each with distinct hydrologic characteristics. The main parameters of the coupled SWAT+*gwflow* model are estimated using parameter estimation software (PEST), and model performance is assessed based on various hydrological metrics. The results are intriguing.

**Response**: We appreciate your feedback! Your comments have been greatly valuable to us, and we've thoroughly reviewed them, making every effort to respond appropriately.

1. Please make sure the abbreviation is defined at the very beginning, GLM, NHD, SWAT, etc.

**Response**: Thank you for your comment. We could include a list of abbreviations at the beginning of the article (after the keywords but before the Introduction). However, we are not sure if this is acceptable to the journal. Therefore, we have elected to define all abbreviations in the first instance they occur in the paper (highlighted in red in the revised version of the article).

2. For table 1, what is the reason for using 1000 m instead of 500 m resolution for Arkansas Headwaters?

**Response**: Thank you for the comment. The grid size of 1000 m is selected for Arkansas Headwaters due to the large size of the watershed (7,940 km$^2$), which leads to a significant increase in computation time if a cell size of 500 m is used. In preliminary simulations, a cell size of 500 m vs. a cell size of 1000 m did not have a noticeable difference on streamflow.

3. Figure 1, please add unit of DEM.

**Response**: Thank you for pointing this out. The DEM unit is added in Figure 1.

4. Figure 7, it looks like the predicted values are smaller than observed value in most cases, what is the reason? It needs more discussion to evaluate the model performance.

**Response**: Thank you for the comment. Yes, generally the PBIAS is positive for all models, indicating an underestimation in streamflow by the model as compared to measured flow. This is not always the case, as there are months (Nanticoke River, Arkansas River, Cache River) in which the model estimates higher streamflow than measured. Underestimation typically occurs for peak flow conditions, which, since the models are also tested for groundwater head, likely is due to inaccurate spatial and temporal representation of significant precipitation events. We believe this is the case with all hydrologic models.

5. Figure 9, why there are only 2 points in the mean absolute error plot?

**Response**: We are not sure what the reviewer is referring to in this comment. Groundwater head results have been evaluated at 128 monitoring wells and are distributed as follows: 7 in the Winnebago River, 26 in the Nanticoke River, 92 in the Cache River, and 3 in the Arkansas Headwaters. The box plot and histogram in Figure 9 display the frequency of the mean absolute error (in meters) of groundwater head for the 128 monitoring wells across the four study watersheds. Please, check line 223.

6. Could you add more detail how you calibrate the model.

**Response**: We appreciate your feedback. The calibration process is explained in Section 2.2.1, which covers Method #1: Parameter ESTimation Tool (PEST) followed by Sensitivity Analysis, and in Section 2.2.2, which discusses Method #2: Iterative Ensemble Smoother (iES) for Parameter Estimation and Uncertainty Analysis (UA).

Technically, the calibration procedure using the Parameter ESTimation (PEST) software for a hydrologic model involves the following steps:

- Model Setup: Configure the hydrologic model to work with PEST. This may include setting up model input and output files in the required formats.
- Parameterization: Recognize the parameters within the hydrologic model that require to be calibrated. These are the parameters that you want PEST to modify to enhance the model's performance.

- Observation Data: Prepare observed data that will be utilized to assess the model's performance during calibration. This data should be in a format that PEST can work with, usually a text file. In our study, we used two watershed responses: streamflow and groundwater head.

- PEST Control File: Create a PEST control file (usually with a .pst extension) that identifies the optimization settings, calibration parameters, observation data, and other relevant information.

- Run PEST: Execute PEST with the control file as input. PEST will interact with your hydrologic model to run simulations with different parameter values.

- Model Simulations: PEST will perform multiple model simulations with several parameter combinations as it gets to minimize the objective function (a measure of the model's fit to observed data of either streamflow or groundwater heads).

The number of iterations in a PEST calibration can vary broadly depending on various aspects, including the complexity of the hydrologic model, the quality and quantity of observed data, the number of parameters being calibrated, and the convergence criteria set by the user. PEST employs an iterative optimization algorithm to adjust model parameters to minimize an objective function (typically a measure of the difference between observed data and model predictions).

Until how many iterations do you consider good calibration?

We have included a text in section 2.2.1. of revised article as follow:

**Lines 234-236**:

*"In our study, we set the maximum number of optimization iterations to 50. However, often PEST converged after 22 iterations (1600 model calls) for Winnebago River, 13 iterations (674 model calls), 36 iterations (2705 model calls) for Arkansas Headwaters, and 13 iterations (843 model calls) for Cache River."*

1. More studies with similar hydrology parameters should be tested to see if the method could have consistent performance.

**Response**: Yes, we agree with the reviewer. We are planning to do many more studies, with other watersheds, in the near future. For this study, we have now included results (calibrated parameters, streamflow statistics, hydrographs, uncertainty plots) for the stand-alone SWAT+ model (i.e., without including the *gwflow* module). These are referenced in the text.